# γ-Secretase cleavage of the Alzheimer risk factor TREM2 is determined by its intrinsic structural dynamics

Andrea Steiner[1,2], Kai Schlepckow[3], Bettina Brunner[3], Harald Steiner[3,4], Christian Haass[3,4,5] & Franz Hagn[1,2,*]

## Abstract

Sequence variants of the microglial expressed TREM2 (triggering receptor expressed on myeloid cells 2) are a major risk factor for late onset Alzheimer's disease. TREM2 requires a stable interaction with DAP12 in the membrane to initiate signaling, which is terminated by TREM2 ectodomain shedding and subsequent intramembrane cleavage by γ-secretase. To understand the structural basis for the specificity of the intramembrane cleavage event, we determined the solution structure of the TREM2 transmembrane helix (TMH). Caused by the presence of a charged amino acid in the membrane region, the TREM2-TMH adopts a kinked structure with increased flexibility. Charge removal leads to TMH stabilization and reduced dynamics, similar to its structure in complex with DAP12. Strikingly, these dynamical features match with the site of the initial γ-secretase cleavage event. These data suggest an unprecedented cleavage mechanism by γ-secretase where flexible TMH regions act as key determinants of substrate cleavage specificity.

**Keywords** dynamics; intramembrane protease; NMR; structure; TREM2
**Subject Categories** Neuroscience; Post-translational Modifications & Proteolysis; Structural Biology
**The EMBO Journal (2020) 39: e104247**

## Introduction

Single transmembrane helices remain a major challenge for structure determination approaches since they are highly flexible and therefore not suitable for crystallization. Solution NMR offers the advantage to study these dynamic domains in different membrane mimetics in solution and to obtain detailed information about their dynamical features. Increasing evidence shows that these hydrophobic membrane spanning helices are more than just membrane anchors but often also involved in transmembrane signal transduction (Matthews *et al*, 2006). Furthermore, intramembrane complex formation with adaptor molecules and homo-oligomerization as well as regulated intramembrane proteolysis are crucial events in the context of intracellular signaling, transcriptional regulation of gene expression, and degradation of membrane proteins, respectively. Regulated intramembrane proteolysis is also key for amyloid β-peptide (Aβ) generation (Lichtenthaler *et al*, 2011), the major component of amyloid plaques invariably found in all Alzheimer's disease (AD) brains (Haass & Selkoe, 2007). Here, the β-amyloid precursor protein (APP) is first cleaved by the β-site APP cleaving enzyme (BACE1) (Vassar *et al*, 2009), which removes the bulk of the ectodomain. γ-Secretase (Wolfe, 2019), then generates Aβ from the membrane-retained APP C-terminal fragment (CTF) by a number of sequential, approximately 3 amino acids spaced cleavages, initiated at the so-called ε-cleavage site located very close to the C-terminal end of the TMH (Steiner *et al*, 2018). Early on, it has been proposed (Beel & Sanders, 2008) that a γ-secretase substrate such as APP has to undergo a helix-to-coil transition to allow the initial cleavage to occur (Sato *et al*, 2009). Numerous type I-oriented γ-secretase substrates have been identified in the past (Haapasalo & Kovacs, 2011; Jurisch-Yaksi *et al*, 2013). Like for APP, sheddases such as BACE1 or members of the ADAM (A Disintegrin and metalloproteinase domain-containing protein) family generate an N-terminally truncated substrate small enough to enter the γ-secretase complex (Struhl & Adachi, 2000; Bolduc *et al*, 2016). However, it is still unclear how γ-secretase selects its substrates, since not every protein with a small enough ectodomain can be cleaved by γ-secretase (Hemming *et al*, 2008). Thus, in addition to membrane orientation and size of the ectodomain other molecular signatures must exist allowing interaction with the γ-secretase complex and recruitment to the active site (Fernandez *et al*, 2016; Fukumori & Steiner, 2016). Unlike soluble proteases, γ-secretase and other intramembrane cleaving enzymes usually do not recognize their substrates

1 Bavarian NMR Center at the Department of Chemistry and Institute for Advanced Study, Technical University of Munich, Garching, Germany
2 Institute of Structural Biology, Helmholtz Zentrum München, Neuherberg, Germany
3 German Center for Neurodegenerative Diseases (DZNE) Munich, Munich, Germany
4 Biomedical Center (BMC), Chair of Metabolic Biochemistry, Faculty of Medicine, Ludwig-Maximilians-Universität München, Munich, Germany
5 Munich Cluster for Systems Neurology (SyNergy), Munich, Germany
*Corresponding author. Tel: +49 89 289 52624; E-mail: franz.hagn@tum.de

via short linear sequence motifs (Beel & Sanders, 2008). Rather, they more likely rely on largely unknown intrinsic structural features of the transmembrane domain which provide flexibility enabling the protease to unfold the transmembrane helix and to cleave the peptide bonds. Indeed, mutations that affect transmembrane domain flexibility in the cleavage site region of APP can markedly alter γ-secretase cleavage (Fernandez *et al*, 2016). Likewise, flexibility via a naturally occurring hinge in the TMH as in the case of APP can also be critical for γ-secretase cleavage (Götz *et al*, 2019). Recent structural studies of γ-secretase in complex with APP or Notch1 substrates suggest that unfolding of the cleavage region is caused by formation of a hybrid β-strand between substrate and enzyme (Yang *et al*, 2019; Zhou *et al*, 2019). Such major structural changes may be facilitated by intrinsic dynamics of the substrate transmembrane domain. Despite these recent insights, it has remained unclear which properties of substrate TMHs determine cleavage specificity.

We here used the triggering receptor expressed on myeloid cells (TREM) 2 as a substrate to investigate whether and how structural flexibility of the transmembrane helix (TMH) could be involved in substrate recognition by γ-secretase. TREM2 is a type I membrane protein, which is exclusively expressed in microglia in the brain (Colonna, 2003). Large genome wide analyses revealed sequence variants specifically within TREM2, which strongly increase the risk for several neurodegenerative diseases (Jonsson & Stefansson, 2013; Jonsson *et al*, 2013; Jiang *et al*, 2016). Apparently, these sequence variants prevent activation of microglia upon neuronal insults, by locking them in a homeostatic state (Keren-Shaul *et al*, 2017; Mazaheri *et al*, 2017). Loss of TREM2 function is associated with reduced proliferation, survival, phagocytosis and lipid sensing of myeloid cells, as well as brain wide reduced energy metabolism and cerebral blood flow (Kleinberger *et al*, 2014; Ulland *et al*, 2015, 2017). TREM2 requires association with the disulfide-linked homo-dimeric signaling partner DNAX activating protein of 12 kDa (DAP12) or DAP10 upon ligand activation for downstream signaling (Hamerman *et al*, 2006; Peng *et al*, 2010). Association of DAP12 with partner proteins is mediated by a salt bridge between their TMHs in the membrane, as previously shown for DAP12 in complex with the natural killer (NK) cell-activating receptor NKG2C (Call *et al*, 2010). In TREM2, a lysine residue is located within the TMH and its homozygous mutation has been connected to various diseases, like Nasu-Hakola disease (Paloneva *et al*, 2002).

TREM2-mediated signaling is terminated by shedding of the extracellular domain by ADAM 10 or 17 producing soluble TREM2 (sTREM2) and a membrane-tethered C-terminal fragment (TREM2-CTF) (Schlepckow *et al*, 2017; Thornton *et al*, 2017), which subsequently undergoes regulated sequential intramembrane proteolysis (Wunderlich *et al*, 2013) (Fig 1A). This degradation process is initiated at the C-terminal end of the TMH at the so-called ε-site. Since γ-secretase activity leads to TREM2-CTF degradation, its inhibition stabilizes the TREM2-CTF:DAP12 complex (Wunderlich *et al*, 2013; Zhong *et al*, 2015). Most likely DAP12 and the TREM2-CTF must dissociate to enable further cleavage by γ-secretase (Wunderlich *et al*, 2013; Zhong *et al*, 2015). Cleavage of the TREM2-CTF by γ-secretase results in the release of the TREM2 intracellular domain (TREM2-ICD) (Lichtenthaler *et al*, 2011; Wunderlich *et al*, 2013) and consequently in the degradation of the membrane-retained CTF.

Here, we set out to characterize the structural and dynamical features of a substrate TMH that are recognized and decoded by γ-secretase, which is still poorly understood so far. By solving the structure of the TREM2-TMH by NMR spectroscopy, we found that a region of lower structural order nearby a membrane-located charged residue is recognized and site-specifically cleaved by γ-secretase. Reduction of the intrinsic dynamics of TREM2-TMH by charge removal results in an altered cleavage site that is located at the C-terminal end of the more stable α-helical secondary structure. Thus, we suggest that TREM2 cleavage specificity by γ-secretase is regulated by an unprecedented mechanism based on the identification of flexible and less ordered regions.

# Results

### TREM2-TMH adopts a charge-induced kinked α-helical structure

For our *in vitro* and structural studies, the TREM2-TMH could be produced in high yields as a fusion protein with GB1 and purified in dodecyl-phosphocholine (DPC) micelles. This yielded a homogenous preparation as judged by SEC and SDS–PAGE (Fig EV1A and B) that was suitable for further structural investigations. Next, using a U-$^2$H, $^{13}$C, $^{15}$N-labeled TREM2-TMH (aa 161–206) sample in 300 mM DPC micelles, NMR backbone resonance assignments were obtained for 88% or all non-proline residues using a set of 3D-triple resonance experiments at 37°C (Fig 1B). Estimation of the secondary structure content using secondary $^{13}$Cα and $^{13}$Cβ chemical shifts yielded an α-helical secondary structure between residues 173 and 198, corresponding to an NMR-derived α-helical content of 57% (Fig 1C), in excellent agreement with the results obtained by CD spectroscopy (Fig 1D). However, the magnitude of the secondary chemical shift indicating α-helix propensity decreases

---

**Figure 1. Structural analysis of TREM2 transmembrane helix (TMH) in dodecyl-phosphocholine (DPC) micelles.**

A  Proteolytic processing of TREM2. First, ADAM10 or 17 cleave off the ectodomain followed by sequential cleavage of the CTF by γ-secretase creating a p3-like extracellular peptide and the intracellular domain (ICD). Only an intact CTF can bind to DAP12. CTF: C-terminal fragment.

B  2D-[$^1$H;$^{15}$N]-TROSY spectrum of 400 μM $^2$H,$^{13}$C,$^{15}$N-labeled TREM2-TMH (residues R161-H206) in DPC micelles at 37°C. Assigned resonances are labeled.

C  Secondary chemical shifts plotted against the residue number indicate an α-helical region between residues 173 and 198. The used TREM2-TMH construct along with the predicted transmembrane helix content (aa 175–195) is also indicated.

D  Circular dichroism (CD) spectrum of TREM2-TMH in DPC micelles indicates an α-helical secondary structure content of ~50%.

E  3D-$^{15}$N-edited NOESY spectra show that the transmembrane helix is interrupted at residues 190-192. Dashed red lines indicate sequential amide NOE contacts that are characteristic for an α-helical secondary structure.

F  Final NMR-derived structural bundle of the 20 lowest-energy structures of TREM2 shows that the C-terminal end of the transmembrane helix can adopt various positions relative to the well-defined TMH. r.m.s.d.: root mean squared deviation.

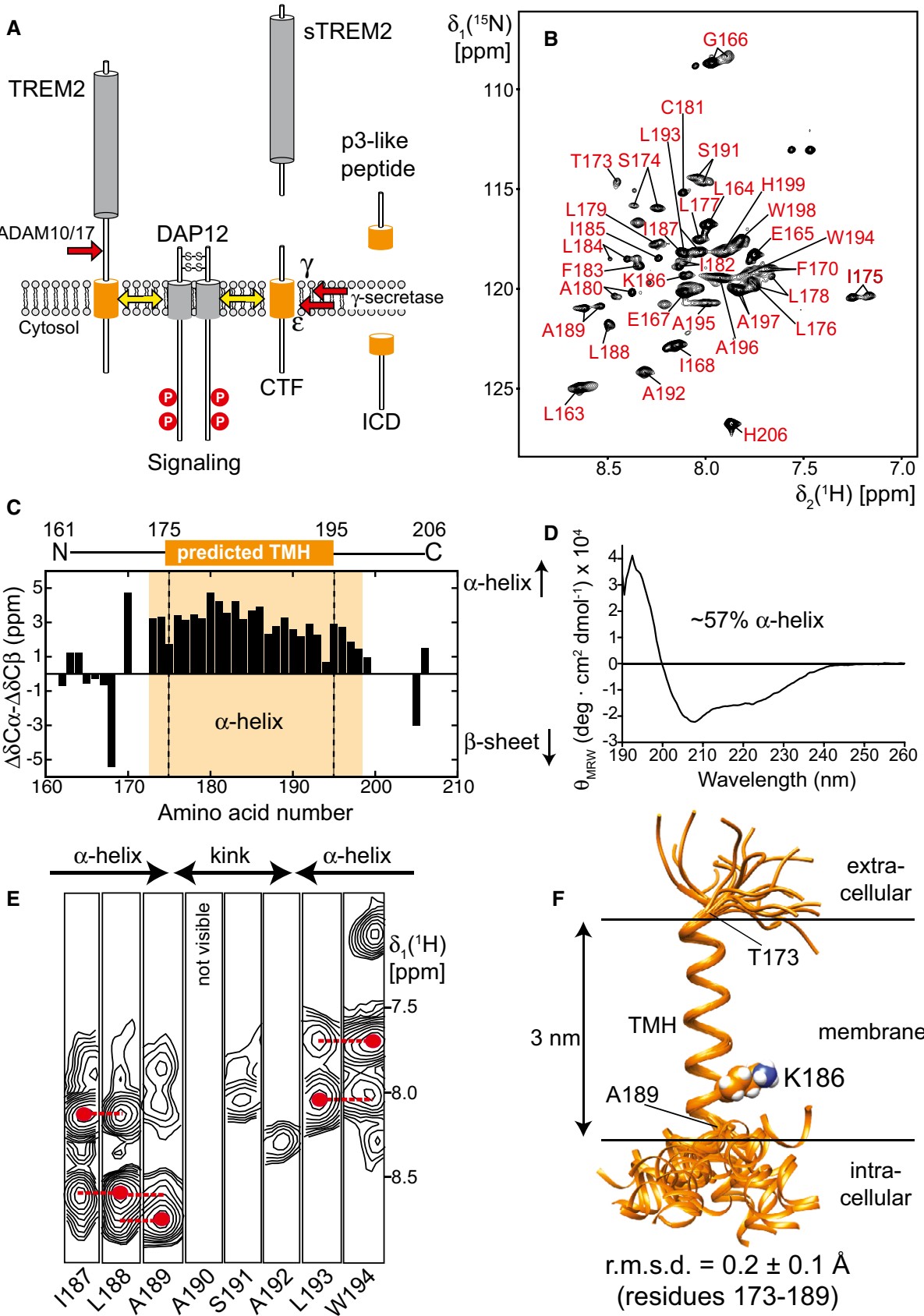

**Figure 1.**

toward the C-terminal end of the helix. Thus, we moved on and determined the structure of TREM2-TMH using nuclear Overhauser effect (NOE) NMR spectroscopy (NOESY). For most parts of the α-helix of TREM2, a typical pattern could be observed in the NOESY where cross-peaks arise from amide protons of sequential neighbors (Fig 1E). However, a 3-residue stretch ranging from A190-A192 at the C-terminal half of the TM helix did not show sequential NOE contacts and exhibited poor signal intensity, indicative of enhanced dynamics and a lack of regular α-helical secondary structure. Thus, in the calculated structure of TREM2-TMH, a pronounced kink in the TMH appears followed by a short C-terminal helix whose orientation relative to the longer N-terminal TMH is variable (Fig 1F). Structural statistics for all investigated TREM2-TMH samples are shown in Table 1. Due to the fact that TREM2-TMH harbors a charged lysine residue at position 186 (K186) that is located in the hydrophobic transmembrane region, we anticipate that this energetically unfavorable location is a key factor that determines its conformation and dynamics. In order to rule out the influence of the non-planar detergent micelle surface, we inserted the TREM2-TMH into di-myristoyl-glycero-phosphocholine/di-myristoyl-glycero-phosphoglycerol (DMPC/DMPG, 3:1) phospholipid nanodiscs assembled with MSP1D1ΔH5 (Hagn *et al*, 2013, 2018; Klöpfer & Hagn, 2019) (Fig EV1C and D) and obtained high-quality 2D NMR spectra that enabled backbone resonance assignments (Fig EV2A), as well as the acquisition of 3D-NOESY spectra (Fig EV2C). Similar to the DPC micelle environment, TREM2-TMH shows the same chemical shift-derived secondary structure content (Fig EV2B) and an identical NOE pattern in lipid nanodiscs, where no contacts can be observed within the amino acid stretch between residues 190 and 192

(Fig EV2C). In line with these observations, a comparison of 2D-[$^1$H,$^{15}$N]-TROSY spectra of TREM2-TMH wt in DPC micelles and DMPC/DMPG nanodiscs shows less pronounced chemical shift perturbations (CSP) as compared to the effect of DAP12 binding (Fig EV2D). The observed spectral changes are most likely caused by the slightly different electronic environment induced by DPC micelles or DMPC/DMPG lipids, and not by changes in the structure of the TMH, as confirmed by NOE contacts. We further investigated the influence of the membrane thickness on the NMR spectra by using nanodiscs assembled with palmitoyl-oleyl-glycero-phosphocholine/palmitoyl-oleyl-glycero-phosphoglycerol (POPC/POPG, 3:1), that forms a thicker membrane as compared to DMPC. As shown in Fig EV2E, the 2D-[$^1$H,$^{15}$N]-TROSY spectra of TREM2-TMH wt in these different nanodisc systems are almost identical, providing further evidence that the structure of TREM2-TMH in DPC micelles represents the biologically relevant conformation.

**TREM2-TMH structure is stabilized by binding to DAP12 or by charge removal**

Next, we used NMR spectroscopy to monitor the complex formation between TREM2-TMH and the dimeric signaling protein DAP12 (Turnbull & Colonna, 2007) as well as the effect of charge removal in TREM2-TMH (K186A) (Fig 2A and B). Upon complex formation, DAP12 induced marked CSPs in the 2D spectrum of $^2$H,$^{15}$N-labeled TREM2-TMH that are located at the C-terminal end of the transmembrane helix, i.e., in the unfolded kink region (Fig 2A and C). Addition of mutant DAP12 lacking its negative charge (D50A) to isotope-labeled TREM2-TMH did not lead to CSPs (Fig EV3A and B),

**Table 1. Structural statistics of TREM2-TMH in DPC micelles.[a]**

| | TREM2-TMH | | |
| --- | --- | --- | --- |
| | **wt** | **K186A** | **DAP12 complex** |
| Structural information | | | |
| Number of NOEs restraints | 64 | 58 | 69 |
| Number of hydrogen bond restraints | 19 | 24 | 24 |
| Number of dihedral angle restraints (TALOS+ (Shen *et al*, 2009)) | 68 | 64 | 66 |
| Backbone rmsd for entire transmembrane helical region (aa 173–198) (Å)[b] | 2.0 ± 0.6 | 0.6 ± 0.2 | 0.7 ± 0.4 |
| Backbone rmsd for first α-helix (aa 173–189) (Å) | 0.2 ± 0.1 | 0.3 ± 0.1 | 0.4 ± 0.2 |
| Ramachandran map analysis[c] | | | |
| Most favored regions | 97.3% | 100% | 81.1% |
| Additionally allowed regions | 0% | 0% | 13.5% |
| Generously allowed regions | 2.7% | 0% | 5.4% |
| Disallowed regions | 0% | 0% | 0.0% |
| Deviation from restraints and idealized geometry | | | |
| Distance restraints (Å) | 0.030 ± 0.003 | 0.023 ± 0.002 | 0.022 ± 0.001 |
| Dihedral angle restraints (deg) | 0.066 ± 0.033 | 0.24 ± 0.02 | 0.33 ± 0.018 |
| Bonds (Å) | 0.0022 ± 0.0001 | 0.0029 ± 0.0003 | 0.0029 ± 0.0001 |
| Angles (deg) | 0.576 ± 0.002 | 0.698 ± 0.007 | 0.679 ± 0.004 |
| Impropers (deg) | 1.1 ± 0.1 | 1.7 ± 0.2 | 2.1 ± 0.1 |

[a]Analysis of the 20 lowest-energy structures.
[b]rmsd values are calculated relative to a non-minimized average structure of each ensemble.
[c]Ramachandran analysis with PROCHECK-NMR (Laskowski *et al*, 1996) was performed on the lowest-energy structure.

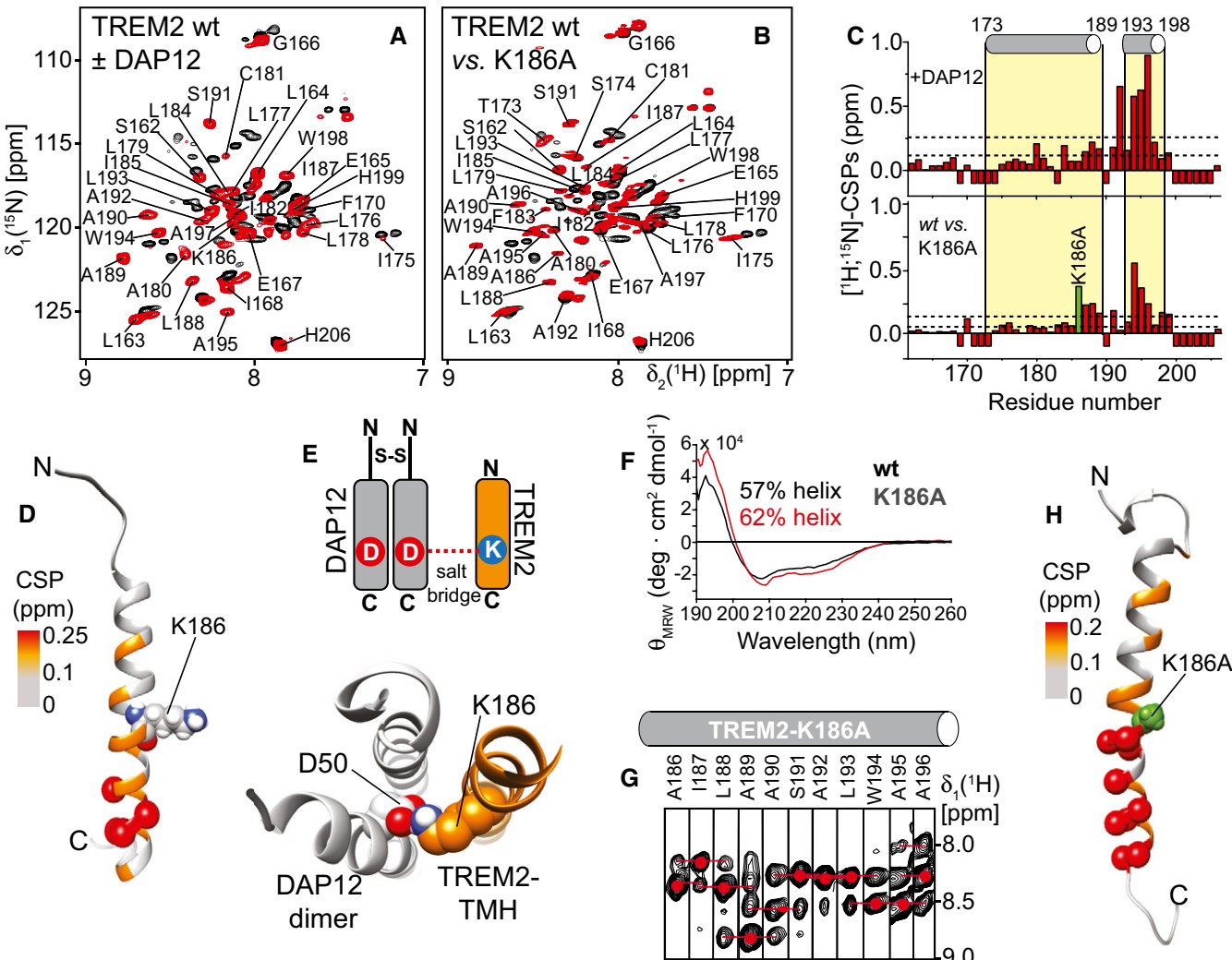

**Figure 2. Interaction of TREM2-TMH with the signaling partner DAP12.**

A, B  Overlay of 2D-[$^1$H,$^{15}$N]-TROSY spectra of $^2$H,$^{15}$N-labeled TREM2-TMH in DPC micelles (black) and in complex with (A) unlabeled DAP12 (red) or (B) TREM2-TMH lacking its positively charged lysine residue (K186A, red).

C  Chemical shift perturbations (CSPs) between the two spectra in panels (A) or (B). The α-helical secondary structure as determined for wild-type TREM2-TMH is indicated above and the position of the K186A mutation is labeled in green.

D  NOE-derived lowest-energy structure of TREM2-TMH in complex with disulfide-bridged dimeric DAP12 shows one continuous α-helix.

E  The experimental structure of TREM2-TMH was docked to a previously determined structure of DAP12 in complex with NKG2C (Call *et al*, 2010). The positive charge in K186 is engaged in a salt bridge with D50 of DAP12. This charge complementarity together with hydrophobic interactions stabilize the complex and lead to a straight α-helical conformation in TREM2-TMH. D50 of the second DAP12 monomer is not in direct contact with TREM2, similar to the DAP12-NKG2C complex (Call *et al*, 2010).

F  For TREM2-K186A-TMH, a ~7% increase in α-helical secondary structure can be detected by CD spectroscopy.

G  A continuous α-helical secondary structure in the kink region (aa 189–192) was probed by sequential amide NOE contacts in TREM2-K186A-TMH. Red dots indicate the chemical shift of intra-residual (i) amide resonance and red broken lines the inter-residual connectivity (i±1).

H  Removal of the positive charge (K186A) leads to a very similar structure of the TREM2-TMH as in its complex with DAP12.

Data information: The CSP values color coded onto the NOE-derived structures in each case (E, H) show similar patterns (color coding according to the dotted lines in panel (C), i.e., lower line: mean CSP value, upper line: mean ± standard deviation). The location of the K186A mutation is indicated by a green sphere. Spheres shown in panels (E and H) represent residues with CSP values above the mean value plus one standard deviation.

confirming that the ionic interaction between K186 in TREM2 and D50 in DAP12 is governing complex formation. Identical results with no detected CSPs were obtained in an NMR titration using isotope-labeled TREM2-TMH K186A with DAP12 wt, further corroborating the crucial role of the salt bridge for complex formation

(Fig EV3C). In order to monitor the structural effects of DAP12 binding on TREM2-TMH, we determined the solution structure of isotope-labeled TREM2-TMH in complex with unlabeled DAP12 using NMR chemical shift information and NOESY experiments. In line with the previous finding that DAP12 binding leads to TREM2-

CTF stabilization (Zhong *et al*, 2015), we here were able to show that this interaction leads to the formation of a continuous α-helix ranging from residues 173 to 198 without the previously kinked region (Fig 2D, Table 1). The structure of the TREM2-TMH in the DAP12-bound conformation was used to obtain a structural model of the complex with DAP12. In the structure of DAP12 in complex with the transmembrane helical region of NKG2C (Call *et al*, 2010), the main interaction between DAP12 and NKG2C is a salt bridge between Asp50 and Lys89. Since DAP12 is a type I and NKG2C a type II transmembrane protein, the interaction takes place in an antiparallel manner in this case. In contrast, TREM2 is a type I transmembrane protein and thus binds to DAP12 in a parallel orientation. Thus, DAP12 appears to be capable of recognizing partner proteins in both orientations. The structural information on the binding site in DAP12 deducted from its complex structure with NKG2C (Call *et al*, 2010) was utilized for NMR data-driven docking calculations of the TREM2-DAP12 complex using the HADDOCK web server (van Zundert *et al*, 2016). For TREM2-TMH, NMR CSPs (Fig 2C, upper panel) were used to define its interface with DAP12. The complex structure that showed the best HADDOCK score and the correct parallel orientation of the two proteins was used for further visualization. In this complex, structural model TREM2 and DAP12 form the expected salt bridge between K186 in TREM2 and D50 in DAP12 (Fig 2E), leading to charge neutralization and TREM2-TMH helix stabilization. Since these studies clearly show that the positive charge within the TREM2-TMH is essential for the interaction with DAP12 and its structural instability, we designed a K186A variant of TREM2-TMH that can serve as a mimic of the neutralized DAP12-bound form. Indeed, a comparison of the 2D-[$^1$H,$^{15}$N]-TROSY spectra of the wild-type versus the K186A variant (Fig 2B) indicates pronounced CSPs. The magnitude and the overall pattern of the CSPs along the protein sequence (Fig 2C, lower panel) are very similar to the CSPs detected upon complex formation with DAP12 (Fig 2C, upper panel). CD spectroscopy with TREM2-TMH K186A indicated an increase in α-helical secondary structure content as compared to the wild-type protein (Fig 2F). Thus, charge removal is most likely resulting in a similar structure of TREM2-TMH as in the complex with DAP12. NOE-based structure determination (Fig 2G, Table 1) of the K186A variant demonstrates that indeed a straight α-helix between residues 173 and 198 is present, without the kink that was observed in the wild-type scenario (Fig 2H).

### TREM2-TMH dynamics is induced by interactions with the lipid headgroups and can be abolished by mutation

Since conformational flexibility plays a crucial role in the recognition of intramembrane protease substrates, we decided to investigate the nanosecond to picosecond dynamics of TREM2-TMH wild-type as well as the K186A variant. For this, we recorded $T_1$, $T_2$ and {$^1$H},$^{15}$N-heteronuclear NOE experiments at 950 and 600 MHz proton frequencies and were able to obtain a generalized order parameter ($S^2$) for all observed backbone amide resonances in each protein variant (Fig 3A and B). $S^2$ can adopt values between zero and one, indicative of unrestricted or completely rigid dynamics, respectively (Lipari & Szabo, 1982a,b). The observed pattern of $S^2$ values along the TREM2-TMH wt sequence correlates very well with the presence of secondary structure elements. $S^2$ values of around 0.9 for the ordered first half of the TMH, of around 0.6

for the unstructured kink region, and 0.7 for the C-terminal helix can be extracted. These data suggest that the first half of the TMH is rigid, the kink region highly dynamic, and the C-terminal helix again more rigid (Fig 3A). The same set of experiments were recorded for the K186A variant. In contrast to the wild-type scenario, the entire amino acid sequence of the α-helical part of TREM2-TMH K186A (residues 173–198) shows order parameters of ~0.9, indicating a rigid and continuous helix without flexible loop regions (Fig 3B). In order to characterize the molecular basis of the observed differences in dynamics, we performed molecular dynamics simulations of TREM2-TMH wt and K186A inserted into a phospholipid bilayer membrane composed of POPC and POPG lipids (identical to the NMR sample conditions used in Fig EV2E), a native-like phospholipid composition. Figure 3C and D show snapshots of the two simulations at various time points, as indicated in each figure. We used a starting structure from the calculated NMR ensemble with linear helix geometry in both cases. During the simulation, the positive charge of the K186 side chain in the wild-type TREM2-TMH is moving toward the phosphate moieties in the phospholipid headgroups leading to a tilt in the helix orientation relative to the membrane surface. Once the K186 side chain is in contact with the phosphate, the amino acids at the C-terminus are pushed outside the hydrophobic region of the bilayer, leading to the experimentally observed kink in helix geometry. However, in order to reproduce the formation of the kink by MD, by far more extended simulation times would be required. Since we have experimental data on the final state of the TMH, we were mostly interested to observe the contact formation between K186 and the headgroups. In contrast, the MD trajectory for the K186A variant is very stable with only minor tilting or translational motions. Due to the missing charge in the latter case the helix is stably inserted in the membrane, in agreement with the observed NMR data.

### Position of TREM2-TMH in the membrane

The MD simulations of wild-type TREM2-TMH suggest that the C-terminal part of the helix is positioned outside the hydrophobic region of the membrane. In order to experimentally prove this finding, we performed paramagnetic relaxation enhancement (PRE) experiments with the TREM2-TMH in DPC micelles using the free radical agent 16-doxyl-stearic acid (16-DSA). This radical leads to NMR line broadening in protein regions that are located in the hydrophobic interior of the detergent micelle, which can readily be observed by a strong decrease in 2D-NMR signal intensity. Figure 4A shows the intensity ratio of the backbone amide resonances in TREM2-TMH in the presence of reduced (radical quenched by ascorbic acid) or oxidized 16-DSA (intact radical) for the wild-type and K186A proteins, respectively. In the wild-type protein, the signal intensity ratio is strongly reduced within the transmembrane region (residues 173–189) but adopts higher values in the kink region and in the C-terminal helix. This pattern suggests that the first helix is indeed a transmembrane helix whereas the kink region and the C-terminal helix are located at the surface of the DPC micelle. For the K186A variant (orange bars in Fig 4A), the observed intensity ratios are markedly altered in comparison with the wild-type protein. In this case, there is a drop in the signal intensity ratio within almost the entire helical secondary structure

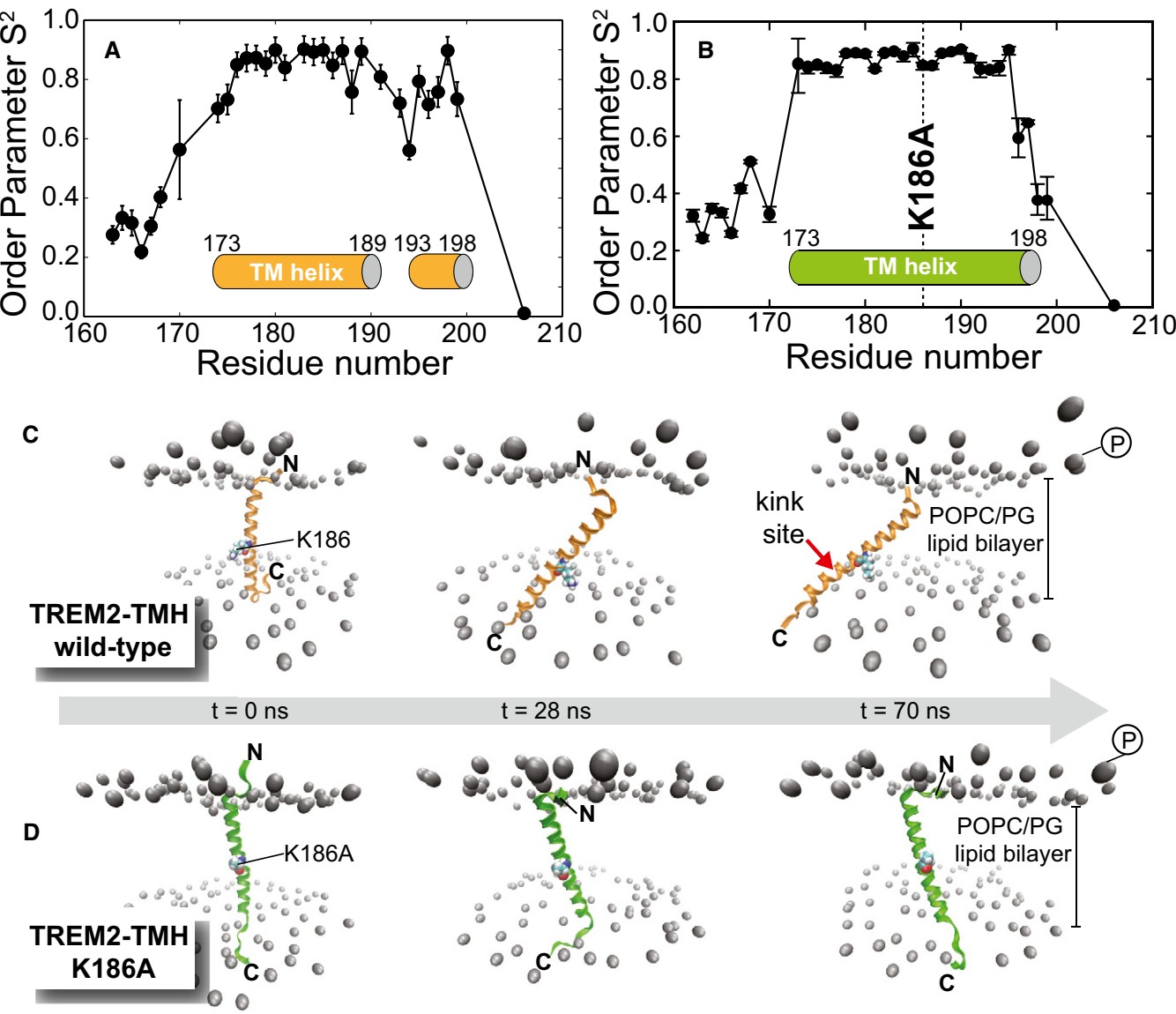

**Figure 3. Dynamics of TREM2-TMH wild-type and the K186A variant.**

A, B  Generalized order parameter $S^2$ for backbone amide resonances derived from NMR relaxation experiments with wild-type TREM2-TMH (A) and the K186A variant (B), indicative of motions in the nanosecond to picosecond time scale. $S^2$ can adopt values between 1 (rigid) and 0 (unrestricted motion). Error bars have been obtained by a Monte-Carlo simulation-based error estimation and motional model selection procedure implemented in the software Relax (Bieri *et al*, 2011) and published previously (Mandel *et al*, 1995).

C, D  Molecular dynamics simulations of TREM2-TMH wt (C) and K186A (D) in a POPC/POPG (3:1) bilayer provide information on the stability of helix insertion into a lipid bilayer membrane. Since K186 is initially located in the hydrophobic membrane, its positively charged side chain is pulled toward the negatively charged phosphates in the lipid head group region leading to a pronounced tilt in helix orientation in the nanosecond time scale and on a μ-second to milli-second time scale to a kink in the C-terminal half of the α-helix, as probed by NMR line broadening. In contrast, in the K186A variant the transmembrane helix is stably inserted within the bilayer.

(residues 173–196), indicative of a stable membrane location. In comparison with the wild-type protein, signal intensity ratios are markedly reduced for the C-terminal half of the TMH. This picture can be corroborated by analysis of NOE contacts (distances of up to 6 Å) between the backbone amide resonances of the two TREM2 variants and selected regions of the DPC molecule (Fig 4B). For the wild-type protein, NOE contacts to the hydrocarbon region of DPC

can be detected for the first helix, confirming that this part is embedded in the membrane. However, these contacts are completely missing for residues in the kink region and can only be faintly observed for the second helix. Furthermore, amino acid residues located in the kink region give rise to a pronounced exchange peak with water, indicating solvent accessibility and a lack of hydrogen bonds in this region. For the K186A variant, NOE contacts to the methyl and

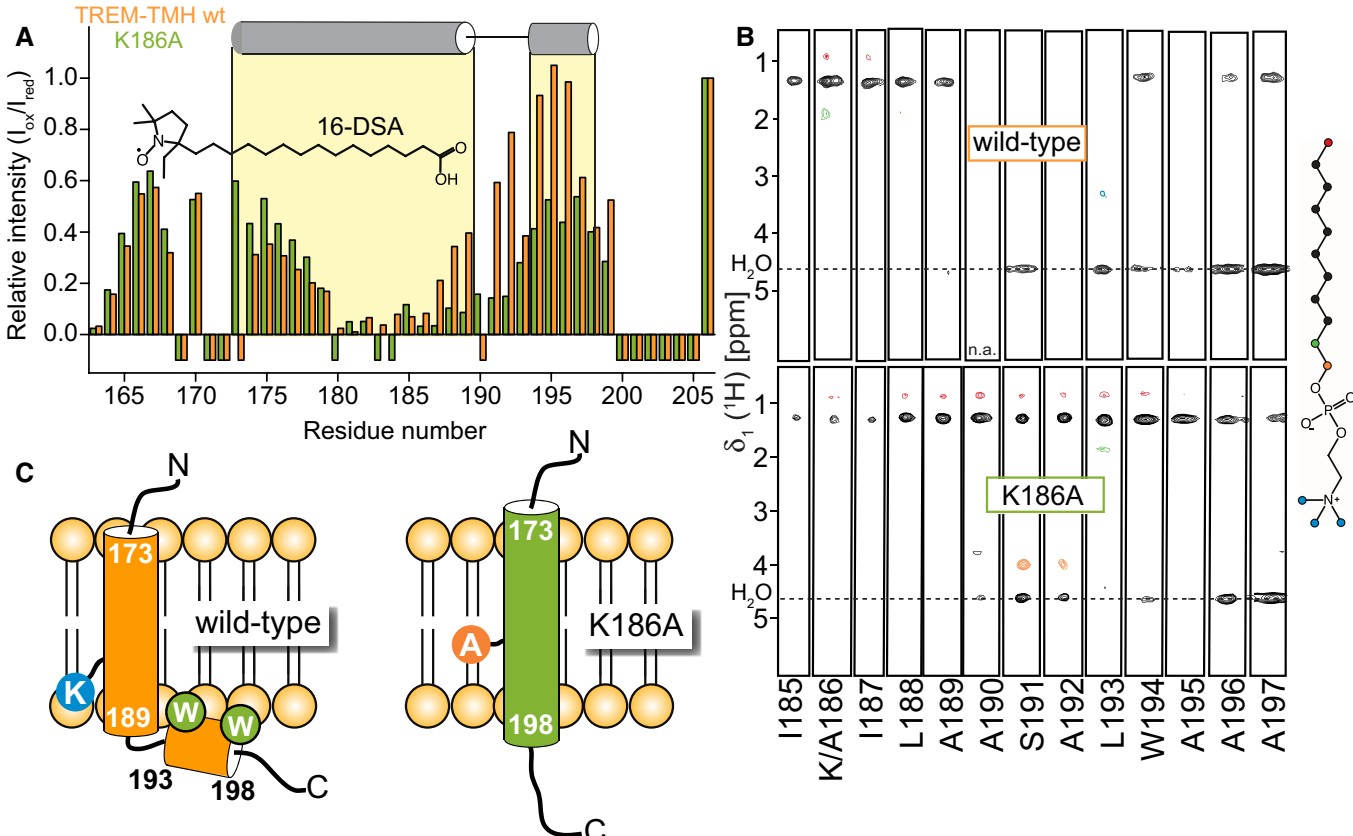

**Figure 4. Membrane position of TREM2-TMH and the K186A variant.**

A Paramagnetic relaxation enhancement (PRE) with a free radical-modified stearic fatty acid in DPC micelles. Low values indicate location inside whereas values close to one suggest a location outside the detergent micelle. Orange bars: TREM2-TMH wt, green bars: K186A variant. The N-terminal residues of TREM2-TMH wt are located deeper inside the micelles as in the K186A case, whereas the C-terminal end is positioned at the outside for wild-type and inside for the K186A variant. Negative bars indicate missing assignments.

B NOE cross-peaks between backbone amides of TREM2-TMH wt (top) or the K186A variant (bottom) and selected regions of DPC (see color coding of the structure of the DPC molecule on the right). The K186A variant shows strong contacts to the hydrophobic aliphatic regions of the detergent throughout the transmembrane helix whereas no such contacts can be seen at the kink site of the wild-type helix, indicating a location outside of the micelle for this part.

C Model of the position of TREM2-TMH wt and K186A within the membrane. The amino acids in the kink region are exposed to the solvent in the wild-type case but buried in the membrane for the K186A variant.

methylene moieties of DPC can be detected throughout the entire helical region, corroborating a transmembrane location of the entire helix. Using these data, a model of the membrane position of both protein variants can be constructed (Fig 4C). TREM2-TMH wt contains a transmembrane region between residues 173 and 189, followed by a kink and an α-helical stretch that is located on the surface of the membrane. The two tryptophan side chains at positions 194 and 198, located on the same side of the C-terminal helix may serve as anchors in the membrane (Fig EV4A), corroborated by NOE contacts between W194 Hε and the detergent. W198 is apparently more dynamic and only loosely attached to the micellar surface leading to line broadening of its Hε resonance. In contrast, TREM2 K186A adopts a straight helical conformation without a kink with both tryptophan side chains inside the membrane (W194) or located at the more solvent exposed headgroup region (W198) (Fig EV4B). The NOE pattern obtained for TREM2-TMH wt in DMPC/DMPG lipid nanodiscs is very similar to the one in DPC micelles (Fig EV4C) with the first α-helix showing contacts to the

hydrocarbon region of the lipids (Fig EV4D), including the terminal methyl groups. In contrast, the C-terminal helix is more loosely attached with contacts to methylene groups of the fatty acid and to the headgroup region of the lipid. Thus, the location in lipids is most likely very similar to the one in detergent micelles (Fig EV4E). In addition, the NOE and PRE data on the membrane location of TREM2-TMH wt and K186A in DPC micelles and in lipid bilayer nanodiscs could be further confirmed by using deuterium exchange experiments, where protein samples have been analyzed by 2D-NMR experiments in $H_2O$ or $D_2O$ buffer. In $D_2O$, all surface exposed or labile amide protons in the protein will exchange to deuterium, giving rise to a loss in signals for these amino acid residues. For TREM2-wt in DPC micelles, almost all signals in a 2D-[$^1H,^{15}N$]-TROSY experiment are lost or reduced in $D_2O$, except those that originate from residues located in the very center of the micelle (Fig EV5A). A similar pattern can be observed in lipid nanodiscs, suggesting that the membrane location is similar in both membrane mimetics (Fig EV5B). In contrast, for the TREM2-TMH K186A

variant the NMR remaining signals in D₂O are more abundant, covering almost the entire TMH (Fig EV5C).

## Intramembrane cleavage of TREM2 by γ-secretase at structurally dynamic regions

To investigate whether intrinsic structural dynamics of the TREM2 TMH determine substrate cleavability, we sought to identify the initial γ-secretase cleavage site of TREM2 releasing the intracellular domain (ICD) by the ε-cleavage. To allow isolation of sufficient amounts of the ICD and to prevent its rapid degradation, we made use of our previously established cell-free *in vitro* γ-secretase assay (Sastre *et al*, 2001). We stably expressed the recombinant TREM2 CTF with its N-terminus at amino acid 158 corresponding to the ADAM10/17 cleavage site (Schlepckow *et al*, 2017; Thornton *et al*, 2017) in human embryonic kidney (HEK) 293 cells. To allow immunoaffinity isolation of the ICD, we added an HA tag to the C-terminus as well as two proline residues to inhibit rapid degradation (Fleck *et al*, 2016) (Fig 5A). Isolated membranes of HEK293 cells stably expressing the TREM2 CTF were incubated for 16 h to allow *de novo* formation of the TREM2 ICD in the presence and absence of the γ-secretase inhibitor L-685,458 (Shearman *et al*, 2000) at 37°C. Immunoprecipitation of the soluble fraction using an anti-HA antibody revealed γ-secretase-dependent generation of the TREM2 ICD (Fig 5B). Immunoprecipitated ICDs were subjected to mass spectrometry as described previously (Fleck *et al*, 2016). This revealed a major peak at 5596,20 Da for wt TREM2, which was absent at 4°C and reduced upon treatment with L-685,458 (Fig 5C). This peak corresponds to a major ε-cleavage site after amino acid 192 within the TMH (Fig 5C–E). Interestingly, the K186A mutation, which strongly stabilizes the α-helical structure of the TREM2 TMH, abolishes cleavage after amino acid 192 and shifts the ε-cleavage to a new site after amino acid 195 and to a lower extent to 193 (Fig 5C–E). Strikingly, the site of intramembrane cleavage of wt TREM2 correlates well with the observed enhanced dynamics between amino acids 189 and 193 (Fig 3A). Furthermore, introduction of the K186A mutation, which reduces structural dynamics within that region (Fig 3B), completely abolished intramembrane proteolysis after amino acid 192 and shifted the cleavage site to the structurally dynamic region at the C-terminus of the stabilized α-helical TMH at position 195 and to a lower extent at position 193. Thus, structurally

dynamic regions appear to be preferred sites for substrate cleavage by γ-secretase in the TREM2 case. To further support this conclusion, we investigated two additional charge mutations, namely K186P and K186L. The presence of proline at position 186 is expected to induce a perturbation in the helical secondary structure, which may resemble the changes in structural dynamics caused by the charged amino acid located normally at this position, whereas the K186L mutation is expected to behave like the K186A mutation. The obtained 2D-NMR spectra of these variants in DPC micelles show that spectral changes occurring for the K186P variant are more pronounced compared with the K186L variant, which is very similar to the K186A case (Fig EV6A–E). Furthermore, secondary structure estimation by CD spectroscopy shows that the α-helical content of the K186P variant is lower than for K186A or K186L but quite similar to the wt protein (Fig EV6F and G). We then performed γ-secretase cleavage assays with these variants. This revealed that the cleavage site of K186L is identical with the one described for the K186A variant (Fig 5C–E), which fits well with our observation that hydrophobic amino acid side chains at position 186 promote formation of a stable TMH. In contrast, the main γ-secretase cleavage site of the K186P variant was between L193 and W194 and therefore shifted only by one amino acid to the C-terminus as compared to the wt protein (Fig 5C–E). Thus, TMH perturbations by either an unfavorable charge in the membrane or a sterically demanding side chain at position 186 lead to similar alterations in the cleavage pattern of γ-secretase. Taken together, these data establish that structural dynamics of TREM2 at the C-terminal end of the TMH determine the site where the initial ε-cleavage takes place.

## Discussion

The aim of this study was to gather structural and dynamical information of the TREM2 TMH and to ask whether and how this could be linked to the initial ε-cleavage by γ-secretase. We employed an interdisciplinary approach combining NMR structural analyses and biophysical methods with membrane-based γ-secretase cleavage assays followed by mass spectrometry to precisely determine the cleavage sites. Owing to a lack in sequence conservation in γ-secretase substrates (Beel & Sanders, 2008), it has been unclear which features allow γ-secretase to select its substrates. In general,

---

**Figure 5.  Identification of γ-secretase cleavage sites in TREM2 wild-type and selected K186 variants.**

A  Schematic overview of the *in vitro* γ-secretase assay to determine the ε-cleavage site(s) of TREM2.

B  Immunoblot detection of ICD accumulation of TREM2 wild-type, K186A, K186P, and K186L variants mediated by γ-secretase cleavage upon overnight incubation of membrane preparations at 37°C. ICD formation is blocked by 1 μM of the transition state analogue L-685,458. Note that inhibition of ICD formation from wt TREM2 was less efficient. Due to the charge within the TMH, wt TREM2 is less efficiently transported to the cell surface (Lanier *et al*, 1998) and therefore retained within the endoplasmic reticulum, where other proteases not related to γ-secretase and therefore not inhabitable by L-685,458 may cleave. *In vitro* generation of the ICD of APP (AICD) was used as a positive control. Immunoblot detection of mature nicastrin and the presenilin 1 NTF, which results from presenilin endoproteolysis necessary to activate γ-secretase (Thinakaran *et al*, 1996), proves comparable amounts of γ-secretase in the individual assays.

C  MALDI MS analysis of the ε-cleavage sites of wt and mutant TREM2 C-terminal fragments. Mass spectrometry reveals ε-cleavage C-terminal of A192 for the wt CTF. MS determination of the ε-cleavage site of TREM2 CTF K186A reveals major and minor ε-cleavage sites C-terminal of A195 and L193, respectively. The exact same cleavage pattern is observed for the K186L mutant. MS analysis of the cleavage pattern of the K186P mutant reveals major and minor ε-cleavage sites C-terminal of L193 and S191. Minor peaks that could not be assigned are indicated by an asterisk.

D  Schematic representation of the ε-cleavage sites of CTF wild-type and selected K186 mutants. Residue 186 is indicated in bold. The location of helices is indicated by boxes.

E  Table listing the calculated and observed masses of both wild-type and mutant ICDs as determined by MALDI MS. Note that molecular weights are increased by 58 Da due to alkylation of C213. As a positive control, we ran immunoblots confirming reduction of APP ICD levels upon γ-secretase inhibition (Appendix Fig S2).

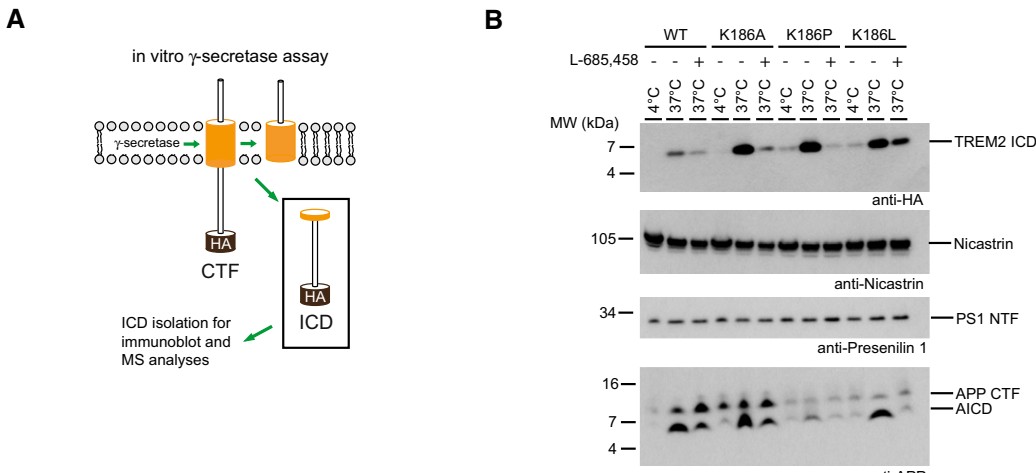

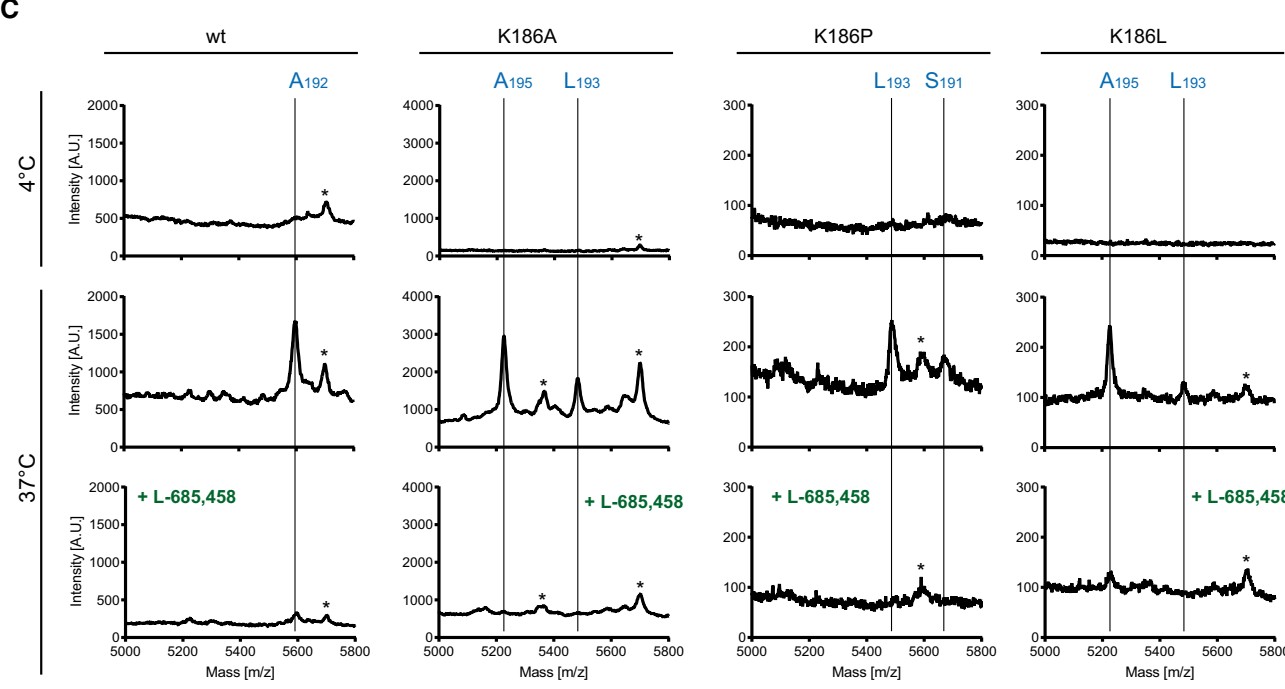

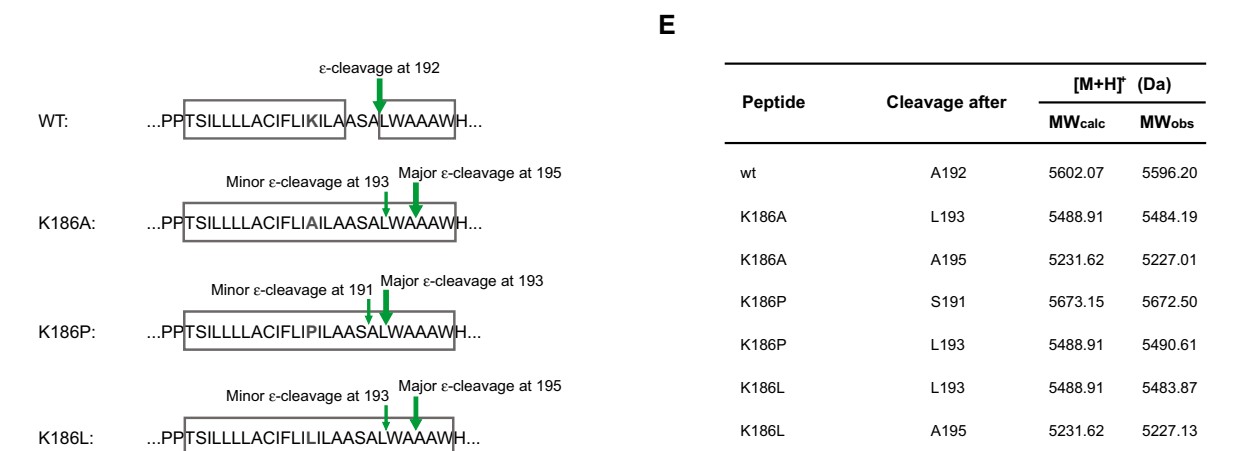

| Peptide | Cleavage after | [M+H]⁺ (Da) | |
|---------|----------------|---------|---------|
| | | MW$_{calc}$ | MW$_{obs}$ |
| wt | A192 | 5602.07 | 5596.20 |
| K186A | L193 | 5488.91 | 5484.19 |
| K186A | A195 | 5231.62 | 5227.01 |
| K186P | S191 | 5673.15 | 5672.50 |
| K186P | L193 | 5488.91 | 5490.61 |
| K186L | L193 | 5488.91 | 5483.87 |
| K186L | A195 | 5231.62 | 5227.13 |

Figure 5.

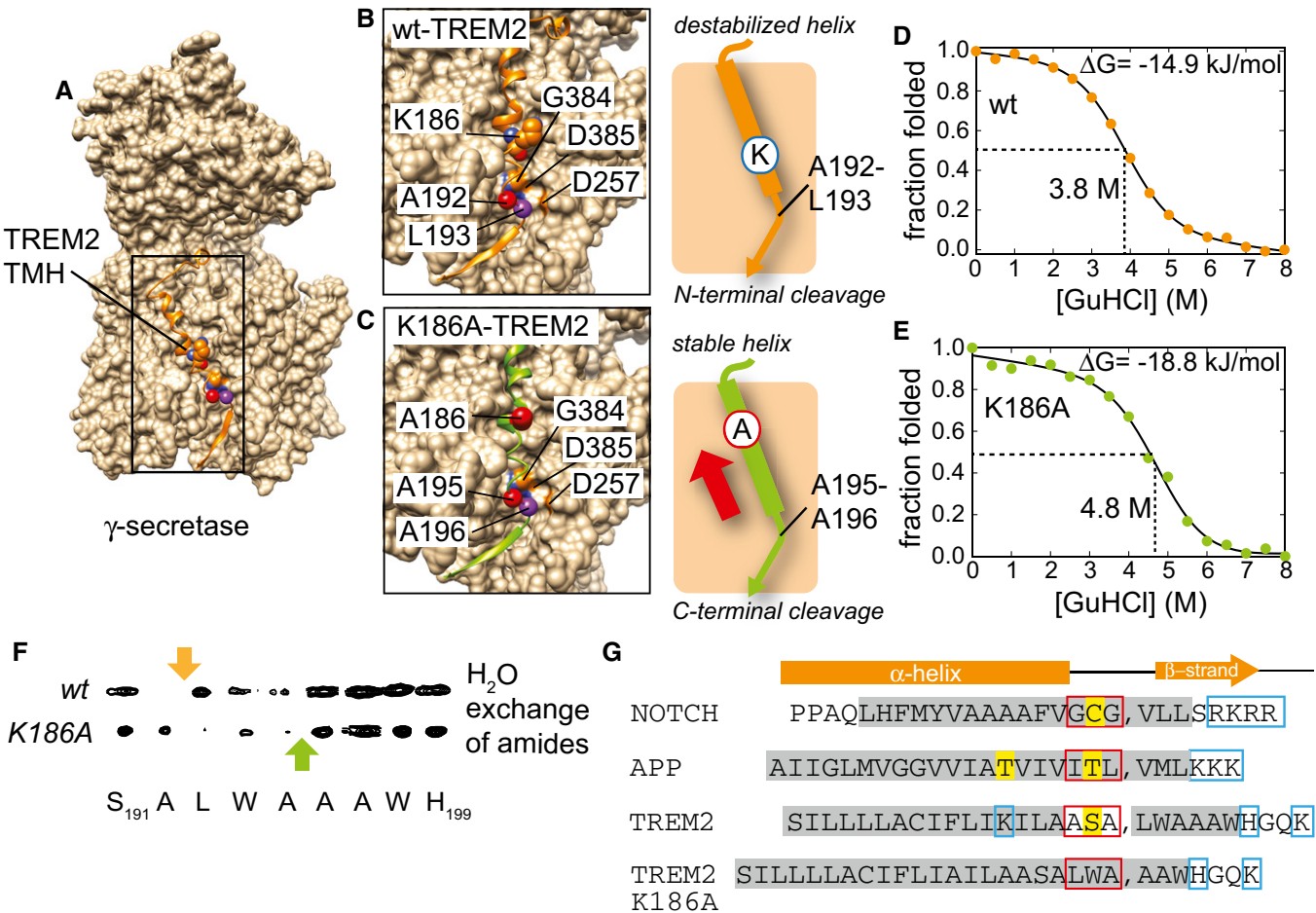

**Figure 6. Cleavage specificity of TREM2-TMH by γ-secretase is governed by its local helix stability.**

A Structural model of the complex of TREM2-TMH wild-type and γ-secretase obtained by docking guided by the cryo-EM structure of the γ-secretase-NOTCH complex (Yang *et al*, 2019). Protein regions in γ-secretase that occlude the substrate-binding site have been removed for clarity.

B, C TREM2-TMH wild-type in the active site of γ-secretase in a register that would lead to cleavage between residues A192 and L193 (marked by spheres in red and purple), as detected by mass spectrometry (see Fig 5). The K186 residue in TREM2 is also shown as well as the catalytically relevant residues in γ-secretase (D257 and D385). In TREM2-TMH K186A, the register of the helix moves upwards leading to positioning of A195 and A196 in the active site (C). The cleavage site needs to be unwound by γ-secretase; thus, the intrinsic stability of the helical substrate is a critical determinant of the cleavage position.

D, E Chemically induced unfolding of TREM2-TMH wild-type (D) and the K186A variant (E) demonstrates that the less dynamic variant is more stable than the wild-type protein.

F Water exchange peak as observed in a 3D-$^{15}$N-edited-[$^1$H,$^1$H]-NOESY experiment which is only absent if the amide moiety is present in a stable hydrogen bond.

G Amino acid sequence comparison of TREM2-TMH wild-type and K186A with Notch1 and APP. α-helical secondary structure elements in the free form of each protein are labeled by gray boxes. Positively charged residues are marked by blue and polar residues close to the cleavage site (indicated by the comma mark) by yellow boxes. Red boxes indicate the unfolded sequence stretches in the bound state preceding the cleavage site. The secondary structure of the substrate helices (or derived from the structural model for TREM2-TMH) in complex with γ-secretase is shown above.

structures of proteases in complex with their substrates show that substrates need to be present in an extended conformation at the protease active site (Madala *et al*, 2010). For γ-secretase substrates, backbone hydrogen bonds at the C-terminal end of their transmembrane helix must be opened to allow the protease active site getting access to their scissile bonds. First insights into how this unfolding of a substrate TMH might occur have only recently been obtained for γ-secretase. Structures of γ-secretase in complex with Notch1 (Yang *et al*, 2019) or APP (Zhou *et al*, 2019) substrate fragments showed that the cleavage region requires to be actively unfolded, which is energetically compensated by the formation of a stabilizing hybrid β-strand composed of a peptide stretch in the substrate

between the cleavage site and the C-terminus as well as regions of the PS1 NTF and CTF in γ-secretase in vicinity to the active site.

Despite these atomic details, the question remains how substrate unfolding is achieved by the enzyme. As shown for the rhomboid protease GlpG and the archaeal presenilin homolog PSH, TMH unfolding can be induced by interaction of the substrate with the protease (Brown *et al*, 2018). For these proteases, the ability of the TMH of the substrate Gurken to adopt a 3$_{10}$-helical conformation is key for the unfolding step when the substrate contacts the protease (Brown *et al*, 2018). Intrinsic substrate TMH properties that provide conformational flexibility such as the diglycine hinge near the middle of the APP TMH have also been shown to play an important

role for cleavage (Götz *et al*, 2019). Although hydrogen-deuterium exchange experiments showed that the cleavage site region in APP is generally stable with respect to local helix flexibility (Pester *et al*, 2013a,b; Scharnagl *et al*, 2014; Götz *et al*, 2019), MD simulations suggested that changes in bending and rotational motions, i.e., in the global flexibility, mediated by this hinge are critical for correct presentation of the cleavage site domain to the active site (Götz *et al*, 2019). Apart from APP, γ-secretase cleavage sites have been determined for many other substrates (Beel & Sanders, 2008), even though experimental details on substrate dynamics are very sparse. Thus, we estimated the dynamics of selected substrate helices by computational methods (using the MemBrain web server (Yin *et al*, 2018)) and found ε-cleavage sites in a region of low predicted order parameter (Appendix Fig S1), consistent with an earlier observation of reduced helicity in this region (Beel & Sanders, 2008).

As determined here for TREM2, an unfolded region manifested by an extreme drop in the order parameter can be intrinsically present in a substrate TMH and oriented toward the outside of the hydrophobic membrane interior. The most critical amino acid residue in TREM2-TMH is the positively charged K186, that is located within the hydrophobic interior of the membrane and which is required for complex formation with DAP12 (Fig 2). Without DAP12, the presence of an unpaired charge in the membrane leads to the formation of a partially unwound helical region with increased conformational dynamics. In order to mimic the DAP12-bound TREM2-TMH conformation, a single K186A point mutation is sufficient for straightening of the TMH while leaving a few C-terminal residues (> A195) slightly more flexible (Fig 3). γ-Secretase cleavage occurs directly in the central unfolded stretch in TREM2-TMH wt, whereas replacement of K186 by alanine shifts the cleavage site to the C-terminal end of the TMH of TREM2, where the stable and rigid α-helical secondary structure of the TMH terminates. Since our *in vitro* cleavage assays have been performed in POPC-rich membrane fractions, whereas the structural information was mainly obtained in DPC micelles, we compared the TREM2-TMH wt spectra in POPC/POPG nanodiscs and DPC micelles (Fig EV2) and could show that the structure of TREM2-TMH is identical in all investigated membrane mimetics. The structure-dependent shift of the γ-secretase cleavage may suggest that the TREM2-TMH, once it has entered the substrate-binding cavity of γ-secretase, may be able to slide vertically in a piston-like movement within the binding groove until it reaches a position where a conformational change, which affects the C-terminal end of the α-helical structure of the TMH, is energetically feasible for the protease (Fig 6A). The resulting register of the substrate within the protease then simply determines which scissile peptide bond is attacked by its active site residues, as sketched for the TREM2-TMH wt and K186A variant in Fig 6B and C. Thus, our data suggest at least in the case of the TREM2 substrate CTF that γ-secretase is able to sample TMHs according to their local stability and inherent flexibility. Such TMH sampling may explain the observation that helix-destabilizing diglycine mutations introduced between the γ- and ε-sites of APP increase the cleavage efficiency and processivity of γ-secretase (Fernandez *et al*, 2016). On the other hand, increased TMH stability can shift the ε-sites to the membrane surface region, as reported for E-cadherin (Marambaud *et al*, 2002).

Using chemical unfolding experiments, we indeed observe that TREM2-TMH K186A is more stable than the wild-type helix

($\Delta G = -18.8$ kJ/mol vs $\Delta G = -14.9$ kJ/mol) (Fig 6D and E). This difference in free energy is in part caused by the presence of 3 additional hydrogen bonds in the K186A variant in line with basic protein folding experiments (Fersht *et al*, 1985). This is in excellent agreement with the continuous α-helical structure and the amide proton exchange characteristics detected by NMR (Fig 6F). A comparison of the amino acid sequences of TREM2-TMH with Notch1 and APP (Fig 6G) shows that the α-helical amino acid stretches located C-terminal of the γ-secretase cleavage site of free APP (Nadezhdin *et al*, 2011) and Notch1 (Deatherage *et al*, 2017) are shorter than for TREM2-TMH wt. These amino acids are succeeded by polar and charged residues that are most likely located outside the membrane. As shown in the structures of the complexes of γ-secretase and APP or Notch1, this structural transition requires local unwinding of around six amino acid residues at the C-terminal end of each TMH, followed by the formation of an intermolecular β-strand with the protease (Yang *et al*, 2019; Zhou *et al*, 2019). The initial energy cost for breakage of these six intra-molecular hydrogen bonds in the substrate TMHs is over-compensated in the complex by the subsequent formation of intermolecular hydrogen bonds (7 for APP, 8 for Notch) between the enzyme and the newly formed C-terminal loop and β-strand of the substrate. A similar behavior may apply to TREM2-TMH K186A, where a continuous helix is present. In contrast, the cleavage site in TREM2-TMH wt is shifted to a more N-terminal position three residues apart. Due to the kink in TREM2-TMH wt, this otherwise energetically less favorable position still comes with a similar energy demand to be invested for local unwinding of its C-terminal helix. These considerations assume that a C-terminal β-strand structure is a general feature of all γ-secretase substrates if bound to the enzyme, which appears to be crucial for energy compensation.

Of particular interest is that the cleavage site in Notch1 is preceded by small and polar residues (GCG motif), further facilitating local unfolding by the enzyme. In APP, a threonine residue is located at the identical position as the cysteine in Notch1 or a serine residue in TREM2. In the APP case, another polar threonine residue is located at the same position inside the membrane as the charged lysine residue in TREM2, which might contribute to its destabilization. Since cleavage at the initial ε-site will lower the overall substrate stability, the energy cost of helix unwinding will be markedly decreased after this first cleavage event. For the substrate APP, this mechanism can explain the multiple downstream cleavage events observed for γ-secretase.

Taken together, as exemplified here for the TREM2-TMH, our study suggests that sampling for intrinsically flexible regions in a TMH is one of the mechanisms allowing γ-secretase to initiate substrate cleavage.

# Materials and Methods

## Sample preparation

The gene of TREM2-TMH (encoding for residues 161–206), TREM2-TMH-K186A, and DAP12-TMH-C33S (residues 21–70) were inserted into in-house-modified pET15b vectors (Merck) and produced in *E. coli* BL21 (DE3) cells as fusion proteins with the protein GB1 harboring an N-terminal His$_6$-tag and a thrombin protease site. Cells

were grown at 37°C and production was induced at an $OD_{600}$ of 0.7 by the addition of 1 mM IPTG. Cells were shaken for 3–4 h at 37°C, harvested by centrifugation at 6,000 *g* and stored at −80°C until further use. To produce [$U$-$^2$H, $^{13}$C, $^{15}$N]-labeled TREM2-TMH, bacteria were grown in M9 medium supplemented with 1 g/l [98% $^{15}$N]-NH$_4$Cl and 2 g/l [98% $^2$H,$^{13}$C]-glucose in 99% D$_2$O (Eurisotop or Sigma-Aldrich). The bacterial pellet was dissolved in lysis buffer (20 mM Tris, pH 8.0, 100 mM NaCl, 1 protease inhibitor tablet (Roche Complete), 0.5 mM EDTA) and incubated with lysozyme for 30 min. After cell breakage by sonication (30 min, 30% amplitude, 10 min total duration) DN*ase*I was added, together with 5 mM MgCl$_2$ and incubated for 30 min on ice. The detergent n-dodecyl-phosphocholine (DPC, Fos-Choline-12) was added to a final concentration of 1.5% (w/v), and the mixture was incubated on a roller shaker for 1.5 h, supplemented with additional 0.9 M NaCl, and incubated for 30 more minutes. Insoluble material was removed by centrifugation (40,000 *g*, 30 min, 4°C), and the supernatant was applied to 5 ml Ni-NTA resin (GE Healthcare, equilibrated with 20 mM Tris, pH 8.0, 100 mM NaCl, 0.1% DPC) using a gravity flow column (Bio-Rad) and incubated with the resin for 1 h at 4°C. After washing with 10 column volumes (CV) with the same buffer + 10 mM imidazole, the protein was eluted using 400 mM Imidazole (5 CV) and loaded onto a HiLoad 16/600 Superdex 200 pg size exclusion column (GE Healthcare). The main peak was collected, supplemented with 25 U thrombin protease and incubated overnight at 37°C. The enzymatic digestion was verified by SDS–PAGE and the protein finally purified using reverse Ni-NTA chromatography. The flow-through was collected and loaded onto a HiLoad 16/600 Superdex 200 column for buffer exchange to NMR buffer (20 mM NaPi, pH 7.0, 50 mM NaCl, 0.5 mM EDTA, 0.1% DPC). The protein concentration was determined by UV/Vis spectroscopy using absorption at 280 nm and a calculated extinction coefficient $\varepsilon_{280}$ of 11,000 M$^{-1}$ cm$^{-1}$. For NMR, the final DPC concentration was adjusted to 200 mM.

### Nanodisc assembly

Nanodisc assembly has been conducted according to previous protocols (Ritchie *et al*, 2009; Hagn *et al*, 2013, 2018). A typical assembly reaction is described below: 1 ml of 600 μM MSP1D1ΔH5 (Hagn *et al*, 2013, 2018), 0.5 ml of a 300 μM solution of $^2$H,$^{15}$N-TREM2-TMH-GB1 in 0.1% DPC, 0.36 ml of a 50 mM DMPC solution in 100 mM cholate, 0.12 ml of a 50 mM DMPG solution in 100 mM cholate were mixed and diluted with 3 ml MSP Buffer (20 mM Tris, pH 7.5, 100 mM NaCl, 0.5 mM EDTA) including 0.05% DPC to a final volume of 5 ml with final concentrations of 120 μM MSP1D1ΔH5, 3.6 mM DMPC, 1.2 mM DMPG, and 30 μM $^2$H,$^{15}$N-TMH-GB1. After incubating the mixture for 1 h at room temperature, the nanodisc assembly reaction was dialyzed twice against 5 l MSP buffer in a 20 kDa cut-off dialysis tube. To yield the loaded nanodiscs, the assembly was added to 3 ml of Ni-NTA resin in a gravity flow column and incubated for 1 h at room temperature while shaking. The Ni-NTA elution fraction was concentrated in a centrifugal device (30 kDa MWCO, Millipore) to a final volume of 1 ml and purified with an ÄKTA Pure system on a S200a column. A homogeneous peak was obtained and concentrated to ~2 ml. 25 units of thrombin were added and incubated overnight at 37°C. The mixture was then purified via reverse Ni-NTA, and the buffer

was changed to NMR buffer (see above) using either a NAP-5 desalting column (GE Healthcare) or by S200a size exclusion chromatography.

### Circular dichroism (CD) spectroscopy

For CD spectroscopy, we used 15 μM of TREM2-TMH and the K186A variant in 20 mM NaPi, pH 7.0, 10 mM NaCl, and 0.1% DPC. Spectral measurements in the range of 190–260 nm were carried out with a Jasco J-715 spectropolarimeter at 20°C in a 1 mm path-length quartz cuvette (Hellma Analytics). The fractional helicity (FH), representing the content of α-helical secondary structure, was calculated using the experimental mean residue weight (MRW) ellipticity at $\lambda = 222$ nm ($\theta^{exp}$) and the corresponding values for the fully helical or fully unfolded conformation ($\theta^h$ and $\theta^u = -36,000$ and $-3,000$ deg cm$^2$ dmol$^{-1}$, respectively) with the following equation: $FH = \frac{\theta_\lambda^{exp} - \theta_\lambda^u}{\theta_\lambda^h - \theta_\lambda^u}$ (Morrisett *et al*, 1973).

### Chemical unfolding experiments

Protein stability was tested by guanidine hydrochloride (GuHCl) induced unfolding with denaturant concentrations up to 8 M. 15 μM TREM2 TMH wt or K186A in 20 mM NaPi, pH 7.0, 10 mM NaCl, and 0.1% DPC was incubated for 30 min with increasing concentrations of GuHCl and the CD ellipticity was measured for a 30-s time interval at a wavelength of 209 nm. The raw data were normalized and fitted with an equation describing a two-state folding model (Privalov, 1979).

### NMR spectroscopy and structure calculation

NMR experiments were recorded at 37°C on Bruker Avance III spectrometers operating at 600, 900, and 950 MHz proton frequency equipped with cryogenic probes. For obtaining backbone resonance assignments and NOE distance restraints, TROSY-based 3D-triple resonance experiments (Salzmann *et al*, 1998) and a 3D-$^{15}$N-edited-[$^1$H,$^1$H]-NOESY experiment were recorded in a non-uniform-sampled (NUS) manner (Hyberts *et al*, 2012a). Schedules were prepared using Poisson-gap sampling (Hyberts *et al*, 2010). 3D-NUS spectra were reconstructed using the iterative soft threshold (IST) method (Hyberts *et al*, 2012b) and processed with NMRPipe (Delaglio *et al*, 1995). Spectral analysis was done with SPARKY4 (Goddard and Kneller, UCSF, 2004) and NMRFAM-SPARKY (Lee *et al*, 2015). Backbone angles for structure calculations were predicted from chemical shift information using the program TALOS+ (Shen *et al*, 2009) and verified by specific NOE contacts. Structure calculations were done with Xplor-NIH (Schwieters *et al*, 2003) using standard protocols. The 20 structures with the lowest restraint violation energies were used to obtain structural statistics of the ensemble. Ramachandran map analysis was performed with PROCHECK (Laskowski *et al*, 1996) via the PDBsum (EMBL-EBI) web server.

### NMR data-driven docking

Docking of the complex between TREM2-TMH and DAP12 was done with the HADDOCK2.2 web server (van Zundert *et al*, 2016) using the herein determined structure of TREM2-TMH in its complex with DAP12 and the structure of DAP12 in complex with the

transmembrane helical region of NKG2C (Call *et al*, 2010). The core residues, in particular D50, that form the complex interface within DAP12 with NKG2C were defined as active (monomer1: V47, D50, T54, A58, V61; monomer2: V42, L43, I46, G49, L53, I57, A60), whereas surface neighbors were defined as inactive residues. For TREM2-TMH, the residues that showed large CSPs in the NMR titration with unlabeled DAP12 were defined as active, and in particular the charged K186 residue (L176, L179, A180, F183, K186, I187, A189, A190, S191, L193, W194, A195, A197). After the calculations, the obtained structural clusters were selected based on the correct orientation of the binding partners and the overall docking score.

The generation of the structural model of the complex between γ-secretase and TREM2-TMH was done by using the γ-secretase-Notch cryo-EM structure (Yang *et al*, 2019) as a template and by replacing the amino acid sequence of Notch by the TREM2 sequence in the program Chimera (Pettersen *et al*, 2004). The exact position of the TREM2-TMH sequence in the binding groove of γ-secretase was defined with the experimentally determined cleavage site for the wild-type and the K186A case.

**Paramagnetic relaxation enhancement (PRE) experiments**

Purified and $^2$H, $^{15}$N-labeled TREM2-TMH and K186A protein samples in 20 mM NaPi, 50 mM NaCl, 0.5 mM EDTA, and 200 mM DPC were mixed with the paramagnetic label 16-doxyl-stearic acid (16-DSA) dissolved in the same buffer. The final 16-DSA concentration was 4 mM which results in approximately one 16-DSA molecule per DPC micelle (DPC aggregation number: ~50). The resulting ratio of protein to 16-DSA was 1:4. 2D-[$^1$H,$^{15}$N]-TROSY spectra were recorded with a recycle delay of 2 s with 96 complex points in the indirect $^{15}$N dimension and 32 scans per increment. The free radical in the spin-labeled fatty acid leads to distance-dependent line broadening of the amide resonances in the protein. The same experimental parameters were used for a reference experiment after the addition of 40 mM ascorbic acid (from a 1M stock in 20 mM NaPi, pH 6.5, 50 mM NaCl, and 0.5 mM EDTA), which is quenching the free radical in the spin label. Signal intensity ratios of the amide resonances in the corresponding spectra can be calculated using the individual peak height values in the oxidized case ($I_{ox}$), i.e., containing a radical, and the reduced spectrum ($I_{red}$), i.e., after reduction of the free radical by ascorbic acid. The resulting values range between 0 and 1 indicating either close or no proximity to the spin-labeled fatty acid hydrocarbon chain, respectively.

**NMR relaxation experiments and data analysis**

$T_1$, $T_2$ relaxation rates as well as {$^1$H}-$^{15}$N-heteronuclear NOEs have been recorded in a fully interleaved manner at two magnetic field strengths (600 and 950 MHz proton frequency). For $T_1$ rates, 14 relaxation delays ranging from 100 ms to 1.8 s have been used, including three duplicates for error estimation. For the determination of $T_2$ rates, we used 14 delays ranging from 16 to 192 ms. Heteronuclear NOE was measured with and without proton saturation (2s saturation time). For all 2D spectra, we recorded 1024 and 96 complex points in the $^1$H and the $^{15}$N dimension, respectively. Order parameters of TREM2-TMH have been extracted with the software Relax (d'Auvergne & Gooley, 2008a,b; Bieri *et al*, 2011) in

fully automated fitting mode using relaxation data at two static magnetic fields (see above).

**Molecular dynamics simulations**

TREM2-TMH was inserted into a hexagonal box of 1-palmitoyl-2-oleoyl-glycero-3-phosphocholine (POPC) and 1-palmitoyl-2-oleoyl-glycero-3-phosphoglycerol (POPG) lipid bilayer (3:1 ratio) using the CHARMM-GUI web server (http://www.charmm-gui.org) (Jo *et al*, 2008) in the presence of 0.15 M KCl. Equilibration of the system was done at 310 K in two phases with 3 cycles each. The force constants to fix the position of the protein and membrane were gradually reduced in each cycle. In the first phase, a timestep of 1 fs and a simulation time of 50 ps in each cycle were used whereas in the second phase a timestep of 2 fs and a simulation time of 200 ps for each cycle were used. Total equilibration time was 750 ps. MD simulations of up to 150 ns duration were carried out with the isothermal-isobaric ensembles at 310 K with the program NAMD (Phillips *et al*, 2005) in the absence of a transmembrane potential. Long-range electrostatic interactions were described using the particle-mesh Ewald method (Darden *et al*, 1993). A smoothing function was applied to truncate short-range electrostatic interactions. Analysis and visualization of the obtained trajectories were done with VMD (Humphrey *et al*, 1996).

**Generation and maintenance of stably transfected HEK293 Flp-In cell lines**

For the determination of the initial ε cleavage within the human TREM2 TMH, we generated a construct that essentially follows the construct design as previously described for neuregulin 1 type III (Fleck *et al*, 2016). In brief, the TREM2 signal peptide at the N-terminus was followed by a FLAG tag, an IM linker sequence, the TREM2 CTF (residues 158-230), a GG linker, a HA tag, and a PP motif for ICD stabilization. The corresponding DNA was purchased from IDT and inserted into the pcDNA5/FRT/TO vector via HindIII (New England Biolabs) and XhoI (Thermo Fisher Scientific) restriction sites. The K186A (AAG>GCG), K186P (AAG -> CCG), and K186L (AAG -> CTG) mutants were generated by site-directed mutagenesis employing PfuTurbo DNA polymerase (Agilent Technologies). All constructs were verified by DNA sequencing (Eurofins Genomics). Transfection of HEK293 Flp-In cells (Thermo Fisher Scientific) with the wild-type and mutant constructs was performed using lipofectamine 2000 according to the manufacturer's recommendations (Thermo Fisher Scientific). Successful transfectants were selected using 100 μg/ml hygromycin B (Thermo Fisher Scientific) and stably transfected cell lines were generated as pools. Culturing of stable cell lines was performed as previously described (Kleinberger *et al*, 2014).

***In vitro* γ-secretase assay**

The *in vitro* γ-secretase assay to generate TREM2 ICDs was essentially performed as previously reported (Sastre *et al*, 2001). Total protein concentrations of cell homogenates of the TREM2 CTF wild-type and mutant cell lines were determined using the BCA assay (Interchim) in order to properly adjust the volume of the buffer (150 mM sodium citrate pH 6.4, 1 mM EDTA, protease inhibitor

mix (Sigma, P8340) used for the assay. To allow generation of the ICD of TREM2, samples were incubated at 37°C overnight and inhibition of γ-secretase-mediated ICD formation was performed using 1 μM L-685,458 (Sigma, L1790). In the last step, soluble (S100) and insoluble (P100) fractions were separated by ultracentrifugation (135,000 *g*; 60 min; 4°C).

### Determination of ε cleavage sites by IP/MS

Soluble fractions as derived from the *in vitro* γ-secretase assay were subjected to peptide alkylation as previously described (Fleck *et al*, 2016) to generate monomeric peptide. In the following, the sixfold volume of IP/MS buffer (0.1% n-octyl glucoside, 10 mM Tris–HCl, pH 8.0, 5 mM EDTA, and 140 mM NaCl) was added and samples were pre-cleared for 1 h using protein G sepharose (GE Healthcare, 17-0618-01). Immunoprecipitation was carried out overnight at 4°C using anti-HA agarose beads (Sigma, A2095). On the next day, beads were washed three times with IP/MS buffer and two times with water. After the last washing step, water was carefully sucked off and beads were stored at −20°C until MS analysis. MALDI MS determination of γ-secretase cleavage sites was performed as previously reported (Fleck *et al*, 2016; Fukumori & Steiner, 2016; Schlepckow *et al*, 2017). Mass spectra were calibrated using the calibration mixture 2 from Sciex, which includes peptides with masses ranging from 1.3 to 5.7 kDa.

### Immunoblotting

The pH of soluble fractions as derived from the *in vitro* γ-secretase assay was adjusted to pH 8.0 using 1 M Tris–HCl, pH 8.0 and EDTA was added to a final concentration of 4 mM. Protease inhibitor mix (Sigma, P8340) was added additionally. The sixfold volume of IP/MS buffer (0.1% n-octyl-glucoside, 10 mM Tris–HCl, pH 8.0, 5 mM EDTA, and 140 mM NaCl) was added, and all samples were subjected to immunoprecipitation overnight at 4°C using anti-HA agarose beads (Sigma, A2095). Samples were washed tow times with IP/MS buffer and two times with water. After the last washing step, water was carefully sucked off and beads were boiled in 2× SDS sample buffer at 95°C for 10 min. For immunoblot analysis, samples were loaded onto precast 10–20% Tris Tricine gels (NOVEX, Thermo Fisher Scientific). Upon gel electrophoresis, proteins were transferred onto a 0.2 μm nitrocellulose membrane (GE Healthcare, 10600001). Membranes were boiled in PBS for 5 min upon protein transfer. TREM2 ICDs were detected using an anti-HA HRP-coupled antibody (clone 3F10, Roche, 1:150). As a positive control, we confirmed ICD generation from APP using a rabbit monoclonal C-terminal anti-APP antibody (Abcam, ab32136, 1:4,000). Comparable amounts of enzymatically active γ-secretase and mature γ-secretase between different assay conditions were confirmed using mouse monoclonal anti-presenilin 1 (Biolegend, SIG-39194, 1 μg/ml) and rabbit polyclonal anti-nicastrin (Sigma, N1660, 1 μg/ml) antibodies, respectively. Samples for anti-presenilin 1 and anti-nicastrin immunoblots were incubated for 10 min at 65°C in Laemmli buffer supplemented with 2M urea before loading onto Tris Tricine gels. The HRP-conjugated goat anti-mouse (1:10,000; Promega) and goat anti-rabbit IgG (1:10,000; Promega) antibodies were used as secondary antibodies. Analysis of immunoblotting was done using enhanced chemiluminescence technique (Pierce).

## Data availability

The NMR resonance assignments and structural coordinates from this publication have been deposited to the BMRB (http://www.bmrb.wisc.edu; assigned accession codes: 50263, 50264, 50265) and PDB (https://www.rcsb.org; assigned accession codes: 6Z0G, 6Z0H, 6Z0I) databanks, respectively.

**Expanded View** for this article is available online.

## Acknowledgements
This work was supported by the Helmholtz Zentrum München and the Helmholtz Society (grant no. VG-NG-1039, to F.H.) and the TUM Institute for Advanced Study funded by the German Excellence Initiative and the European Union Seventh Framework Program (Grant No. 291763, to F.H.). C.H. and H.S. gratefully acknowledge support from the DFG-funded FOR2290. The authors gratefully acknowledge the Gauss Centre for Supercomputing e.V. (www.gauss-centre.eu) for funding this project by providing computing time on the GCS Supercomputer SuperMUC at Leibniz Supercomputing Centre (www.lrz.de). Open access funding enabled and organized by Projekt DEAL.

## Author contributions
AS, KS, BB, and FH performed experiments and analyzed data. AS, KS, HS, CH, and FH designed research and contributed to data interpretation. AS, KS, HS, CH, and FH wrote the paper.

## Conflict of interest
CH and KS collaborate with DENALI Therapeutics. C.H. participated on one advisory board meeting of Biogen and is chief advisor of ISAR Bioscience.

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
