## [Review Process File · The EMBO Journal]

γ -secretase cleavage of the Alzheimer risk factor TREM2 is determined by its intrinsic structural dynamics

Andrea Steiner, Kai Schlepckow, Bettina Brunner, Harald Steiner, Christian Haass, and Franz Hagn

DOI: 10.15252/emboj.2019104247

Corresponding author(s): Franz Hagn (franz.hagn@tum.de)

Review Timeline:

Submission Date:	11th Dec 19
Editorial Decision:	7th Jan 20
Revision Received:	10th Jul 20
Editorial Decision:	24th Jul 20
Revision Received:	28th Jul 20
Accepted:	29th Jul 20

Editor: Hartmut Vodermaier

Transaction Report:

Dr. Franz Hagn
TUM
Department of Chemistry
Lichtenbergstrasse 4
Garching, Bavaria 85747
Germany

7th Jan 2020

Re: EMBOJ-2019-104247
Intramembrane cleavage of TREM2 is determined by its intrinsic structural dynamics

Dear Franz,

Thank you for submitting your manuscript on TREM2 structural dynamics and gamma-secretase substrate selection for our editorial consideration. It has now been assessed by three expert referees, whose comments are copied below for your information. As you will see, all referees acknowledge the importance of the subject and the potential interest of your findings, as well as the quality of the data and their presentation. Pending satisfactory revision of a number of specific concerns raised in the reports, we would therefore be happy to consider the study further for publication as an EMBO Journal article. Particularly important in this respect would be the inclusion of additional functional data, as requested by referee 2 (point 1), and clarifying the issues of possible effects of membrane thickness (referee 3) and of altered substrate stability (referee 2 point 3).

When revising the manuscript, please also include a 'data availability' section for the structural data, as stipulated in our author guidelines.

I should add that it is our policy to allow only a single round of (major) revision, making it important to comprehensively answer to all points raised at this stage. Should you have any specific questions/comments regarding the referee reports and this decision, please do not hesitate to get back to me already during the early stages of your revision.

Further information on preparing and uploading a revised manuscript can be found below and in our Guide to Authors.

Thank you again for the opportunity to consider this work for The EMBO Journal! I look forward to your revision.

With kind regards and best wishes for the new year,

Hartmut

Hartmut Vodermaier, PhD
Senior Editor / The EMBO Journal
h.vodermaier@embojournal.org

- a point-by-point response to the referees' comments, with a detailed description of the changes made (as a word file).
- a word file of the manuscript text.
- individual production quality figure files (one file per figure)
- a complete author checklist, which you can download from our author guidelines (<https://www.embopress.org/page/journal/14602075/authorguide>).
- Expanded View files, replacing Supplementary Information (Please see <https://www.embopress.org/page/journal/14602075/authorguide#expandedview>)

Further information is available in our Guide For Authors:

Revision to The EMBO Journal should be submitted online within 90 days, unless an extension has been requested and approved by the editor; please click on the link below to submit the revision online before 6th Apr 2020:

Link Not Available

Referee #1:

The manuscript by Steiner et al. describes solution NMR studies on a g-secretase substrate (TREM2) to understand the requirements for cleavage. This is an important question in the field of intramembrane proteolysis and is particularly relevant for the mechanism of the g-secretase complex in order to establish the structural requirements that distinguish substrates from non-

substrates.

The basic conclusion of the manuscript is that the C-terminal end of the helical transmembrane domain needs to be flexible or unraveled at the epsilon cleavage site in order for the substrate to be processed. Mutations or complex formation that shift this unfolded region toward helical secondary structure will impede processing/cleavage. This conclusion, which is described as "unprecedented" in the Abstract, is basically the same as that reached in 2009 based on NMR, mutational and g-secretase processing studies of the APP and Notch TM domain substrates (Sato et al. PNAS 106, 1421-1426). This reference was apparently missed by the authors.

Having said that, the current manuscript is exceptionally well crafted on many levels. The first level is that it is comparative in nature. The authors compare the TM helix of TREM2 (a g-secretase substrate), with that of a mutant TREM2 TM domain (K186A) and with the TREM2 helix in complex with DAP12. Both the TM mutation and formation of the complex impede processing, and the authors show that this correlates with a decrease in dynamics of the C-terminal end of the TM domain. Second, the NMR measurements were compared in detergent micelles and membrane nanodiscs. Third, the NMR measurements of dynamics are complemented by MD studies and processing measurements to provide a complete analysis of the correlation of structure and function. On top of that the manuscript is well written and the figures are well-composed to very easily follow the logic underlying the experiments and conclusions drawn. The manuscript very clearly describes for the wider readership of EMBO J. one of the key elements that distinguish substrates from non-substrates by gamma-secretase.

The one (minor) experiment that I was expecting to see was the indole 1H-15N HSQC showing water accessibility (easily done with D2O) to establish the relative positions of W194 and W198 in the micelle. The slices in Figure EV4 - while indicative of location - are not as easy to compare - in my opinion - as simply observing the influence on the HSQC NH resonances in a single spectrum.

Referee #2:

In the presented manuscript Steiner and colleagues analyze the structure of the transmembrane domain of TREM2 (triggering receptor expressed on myeloid cells 2) by NMR, biophysical methods and molecular dynamics (MD), and explore the implications of its intrinsic dynamics for the gamma-secretase mediated endopeptidase cleavage.

The NMR structural data of the TREM2 transmembrane helix (TMH) reveal a region of lower structural order nearby the charged Lys186 residue and convincingly demonstrate that charge removal (Lys to Ala, K186A) markedly reduces the dynamics of TMH and promotes the stabilization of a transmembrane alpha helical structure (residues 174-198).

Interestingly, functional analyses show that altered structural dynamics in the TREM2-TMH results in an altered endopeptidase gamma secretase cleavage site. Specifically, a major endopeptidase cleavage after amino acid 192 within the wild type TREM2 TMH (Fig 5c, e, f) occurs after amino acid 195 (and to a lower extent to 193 (Fig 5d-f)) in the mutant K186A TREM2 substrate.

In the last part of the manuscript, based on the novel data and previous high resolution structural data for gamma-secretase/substrate complexes, the authors suggest that gamma-secretase is able to sample substrate TMHs according to their local stability and inherent flexibility (presented in Figure 6).

This reviewer finds the structural data highly interesting and novel, fitting to the broader view (summarized in the manuscript) where dynamic features of the substrate and the enzyme are

determinant for γ -secretase proteolysis (Fernandez et al., 2016; Szaruga et al., 2017; Götz, Högel et al., 2019; Yang, Zhou et al., 2019, Zhou, Yang et al., 2019). However, this reviewer finds that the functional correlations, validating the structural observations, remain preliminary and require further investigation in order to draw solid conclusions.

Major points:

1) The authors propose a model where "sampling for intrinsically flexible regions in a TMH is one of the mechanisms allowing gamma-secretase to initiate substrate cleavage."

The proposed model is highly interesting, however must be supported by functional data. In the current version of the manuscript, the functional evaluation is very limited and compromises the relevance of the studies. The authors should evaluate the effects of the K186A mutation on gamma secretase kinetics to answer whether the substrate dynamics affect substrate recognition (K_m)? or rate by facilitating helix unwinding?

Furthermore, it would be important to demonstrate that at least another mutation affecting the dynamics of the TREM-2 TMH (such as the K186P) has similar functional effects on the gamma secretase endopeptidase. Otherwise, the authors cannot rule out in their model another specific role of the lysine residue at the position 186 of TREM2. Generating the above described data is feasible, will help significantly with the interpretation of the structural data and support better the model presented in the manuscript.

2) Interestingly, MD simulations and paramagnetic relaxation enhancement (PRE) experiments of wild-type TREM2-TMH indicate that the C-terminal part of the predicted TMH is positioned outside the membrane. In contrast the drop in the intensity ratio within the entire K186A mutant TMH (residues 173-196) indicate a stable membrane-embedded structure. The observation that the part of the substrate that is actually cut by γ -secretase is not embedded in the membrane is unprecedented, the origin is well documented and discussed in the manuscript but the implications for the γ -secretase substrate recognition/cleavage mechanisms are not discussed. The authors should elaborate on the potential implications of this finding.

3) The chemically induced unfolding of TREM2-TMH wild-type (Fig. 6d) and the K186A variant (Fig. 6e) shows that the less dynamic variant is more stable than the wild-type protein. This reviewer wonders if the different stabilities of the wild type vs. mutant substrate have an impact on the stability of gamma-secretase-substrate interactions. It has been recently shown that AD-linked mutations impair the stability of gamma-secretase-substrate interactions and the effect shifts amyloid beta product length (Szaruga et al, Cell 2017). It would be very interesting to analyze how the substrate stability affects the stability of the gamma-secretase-TREM2 complex.

Minor points:

1) Several references do not refer to the original research papers but to review articles. For example:

' γ -Secretase (Wolfe, 2019), then generates A β from the membrane-retained APP C-terminal fragment (CTF) by a number of sequential, approximately 3 amino acids spaced cleavages, initiated at the so-called γ -cleavage site located very close to the C-terminal end of the TMH (Steiner, Fukumori et al., 2018).'

To improve clarity, this reviewer suggests to refer to the original articles or indicate by 'reviewed/discussed in' that the used reference is a review article.

2) Spaces are missing in various parts of the manuscript. Here some examples:

'which subsequently undergoes regulated sequential intramembrane proteolysis(Wunderlich, Glebov et al., 2013) (Fig 1a).'

'and Notch1 (Deatherage, Lu et al., 2017)are shorter than for TREM2.'

'1 mL of 600 μ M MSP1D1 Δ H5(Hagn et al., 2013, Hagn et al., 2018)'

'describing a two-state folding model(Privalov, 1979).'

3) Referencing problem:

(Ref (Morrisett, David et al., 1973))

4) In the introduction a period is missing after: "Complete degradation of the CTF is crucial to prevent inhibition of DAP12 that would encompass productive full-length-dependent TREM2 signaling. (Wunderlich et al., 2013) " .

Referee #3:

In this paper NMR is used to determine the solution structure of a transmembrane substrate of the γ -secretase, TREM2. Variations in the TREM place a person in a higher risk category for neurodegenerative disease. Typically, substrates of intramembrane proteases consist of transmembrane segments, which must be unwound prior to cleavage. For γ -secretase over 80 substrates exist. Recent structures demonstrate that substrates unwind and form β -sheet with the enzyme. It is not known how γ -secretase selects its substrates and the authors propose intrinsic dynamics of the substrate may influence recognitions. This study set out to address this by studying the triggering receptor expressed on myeloid cells (TREM) 2 as a substrate and examination of its structure and cleavage.

The NMR solution structure revealed a helix segment with a less ordered region near a charged Lysine residue, in other words a kink followed by a helical section that was proposed to influence dynamics. To eliminate the possibility of the influence of detergent, the structure was confirmed with the membrane protein in nanodisks. Membrane thickness was also ruled out in influencing this kink. The helical structure was stabilized upon interaction with its physiological partner DAP12. Mutation of this charge residue to Alanine resulted in changes in the chemical shift, suggestive of a continuous helix, with similar chemical shifts (i.e. stabilization) when in association to DAP12. A structural model was generated to show TREM and DAP interact with each other via a slat-bridge (utilizing the Lys).

NOEs were obtained to measure dynamics and the K186A mutation resulted in less dynamics in this region.

MD simulations in lipid bilayers suggest the Lys residues causes the helix to tilt in the membrane via associations with the phosphate headgroups. Paramagnetic relaxation enhancement (PRE) experiments were used to show the C-terminal region is outside of the hydrophobic area, while the K186A is entirely in the micelle. This was conducted in DPC micelles. A cell-free in vitro γ -secretase assay was used to measure release of the C-terminal fragment. They show this using this assay in conjunction with mass spectrometry that the mutation K186A results in a shift in cleavage. A structural model for the different recognition is provided. A novel hypothesis is put forth based on the fact that dynamic regions is one mechanism that influences substrate recognition for γ secretase.

Overall this is a very well written manuscript. The concept that structural dynamic regions play a role in substrate recognition for γ -secretase substrate is interesting given that there is no sequence conservation for the numerous substrates. The data present in the manuscript is

convincing that changes in the single Lys residue alter the structural nature and cleavage of the TREM12 substrate.

Major points:

Is the Lys involved in a salt bridge should its mutation affect interaction with DAP12? We see that mutation of the Asp residue in DAP leads to a loss in chemical shift upon interaction. Did the L186A also have the same effect? This is clear from the figures but the results section does not make this point clearly. i.e. on Page 7.

On page 10 the following statement: This pattern suggests that the first helix is indeed a transmembrane helix whereas the kink region and the C-terminal helix are located outside the membrane, should be adjusted to state In DPC micelles. There is no evidence that this is outside in the physiological cellular environment which would have thicker membranes compared to DPC.

The hypothesis that dynamic flexibility is one mechanism influencing substrate is interesting however in the discussion I would like to see if this is the case for any of the other 80 plus known substrates for g-secretase. Only APP HDX and simulation experiments are discussed.

Minor points to address:

1. Sigma protease inhibitor mix is mentioned several times in the methods, however several exist at Sigma. Please specify precise product, especially since the research is protease focused.

Dear Hartmut,

Thank you for handling our submission and your positive comments and helpful suggestions. After some delays caused by the Covid-19 crisis, we now have been able to compile a revised version of our manuscript. Therein, we provide a large set of new data that address most of the reviewers' concerns. In addition, we have attached a detailed point-by-point response to all reviewer's comments (see below). We are highly grateful for the helpful comments of all three referees and sincerely hope that the improved manuscript is now suitable for publication in *EMBO J*.

Sincerely,
Franz

As you will see, all referees acknowledge the importance of the subject and the potential interest of your findings, as well as the quality of the data and their presentation. Pending satisfactory revision of a number of specific concerns raised in the reports, we would therefore be happy to consider the study further for publication as an EMBO Journal article. Particularly important in this respect would be the inclusion of additional functional data, as requested by referee 2 (point 1), and clarifying the issues of possible effects of membrane thickness (referee 3) and of altered substrate stability (referee 2 point 3).

A: In the revised version of the manuscript we included additional data that specifically address the main concerns of all reviewers. Please see the specific comments below.

When revising the manuscript, please also include a 'data availability' section for the structural data, as stipulated in our author guidelines.

A: This has been updated. We have now deposited the NMR chemical shift information as well as the structural coordinates of the TREM2-TMH wt, K186A and TREM2-TMH in complex with DAP12 in the BMRB and PDB databanks and included all accession codes in the manuscript together with a statement indicating the associated databank accession codes.

Referee #1:

The manuscript by Steiner et al. describes solution NMR studies on a g-secretase substrate (TREM2) to understand the requirements for cleavage. This is an important question in the field of intramembrane proteolysis and is particularly relevant for the mechanism of the g-secretase complex in order to establish the structural requirements that distinguish substrates from non-substrates.

A: Thanks to the reviewer for this positive comment.

The basic conclusion of the manuscript is that the C-terminal end of the helical transmembrane domain needs to be flexible or unraveled at the epsilon cleavage site in order for the substrate to be processed. Mutations or complex formation that shift this unfolded region toward helical secondary structure will impede processing/cleavage. This conclusion, which is described as "unprecedented" in the Abstract, is basically the same as that reached in 2009 based on NMR, mutational and g-secretase processing studies of the APP and Notch TM domain substrates (Sato et al. PNAS 106, 1421-1426). This reference was apparently missed by the authors.

A: We included the mentioned reference. Thanks for this comment.

Having said that, the current manuscript is exceptionally well crafted on many levels. The first level is that it is comparative in nature. The authors compare the TM helix of TREM2 (a g-secretase substrate), with that of a mutant TREM2 TM domain (K186A) and with the TREM2 helix in complex with DAP12. Both the TM mutation and formation of the complex impede processing, and the authors show that this

correlates with a decrease in dynamics of the C-terminal end of the TM domain. Second, the NMR measurements were compared in detergent micelles and membrane nanodiscs. Third, the NMR measurements of dynamics are complemented by MD studies and processing measurements to provide a complete analysis of the correlation of structure and function. On top of that the manuscript is well written and the figures are well-composed to very easily follow the logic underlying the experiments and conclusions drawn. The manuscript very clearly describes for the wider readership of EMBO J. one of the key elements that distinguish substrates from non-substrates by gamma-secretase.

A: Thanks for highlighting the key message and the relevance of our manuscript!

The one (minor) experiment that I was expecting to see was the indole 1H-15N HSQC showing water accessibility (easily done with D2O) to establish the relative positions of W194 and W198 in the micelle. The slices in Figure EV4 - while indicative of location - are not as easy to compare - in my opinion - as simply observing the influence on the HSQC NH resonances in a single spectrum.

A: We now included a spectrum of TREM2-TMH wt and variants in D2O and could confirm the results obtained with NOESY and PRE NMR experiments. These data clearly show that the transmembrane helix is less stable in the wild-type than in the charge-deficient variant. This new data has been included in the new Fig. EV 6.

Referee #2:

In the presented manuscript Steiner and colleagues analyze the structure of the transmembrane domain of TREM2 (triggering receptor expressed on myeloid cells 2) by NMR, biophysical methods and molecular dynamics (MD), and explore the implications of its intrinsic dynamics for the gamma-secretase mediated endopeptidase cleavage.

The NMR structural data of the TREM2 transmembrane helix (TMH) reveal a region of lower structural order nearby the charged Lys186 residue and convincingly demonstrate that charge removal (Lys to Ala, K186A) markedly reduces the dynamics of TMH and promotes the stabilization of a transmembrane alpha helical structure (residues 174-198).

Interestingly, functional analyses show that altered structural dynamics in the TREM2-TMH results in an altered endopeptidase gamma secretase cleavage site. Specifically, a major endopeptidase cleavage after amino acid 192 within the wild type TREM2 TMH (Fig 5c, e, f) occurs after amino acid 195 (and to a lower extent to 193 (Fig 5d-f) in the mutant K186A TREM2 substrate.

In the last part of the manuscript, based on the novel data and previous high resolution structural data for gamma-secretase/substrate complexes, the authors suggest that gamma-secretase is able to sample substrate TMHs according to their local stability and inherent flexibility (presented in Figure 6).

This reviewer finds the structural data highly interesting and novel, fitting to the broader view (summarized in the manuscript) where dynamic features of the substrate and the enzyme are determinant for γ -secretase proteolysis (Fernandez et al., 2016; Szaruga et al., 2017; Götz, Högel et al., 2019; Yang, Zhou et al., 2019, Zhou, Yang et al., 2019). However, this reviewer finds that the functional correlations, validating the structural observations, remain preliminary and require further investigation in order to draw solid conclusions.

A: Thank you for these encouraging comments. In our revised manuscript we tried to address all concerns of this reviewer.

Major points:

1) The authors propose a model where "sampling for intrinsically flexible regions in a TMH is one of the mechanisms allowing gamma-secretase to initiate substrate cleavage."

The proposed model is highly interesting, however must be supported by functional data. In the current version of the manuscript, the functional evaluation is very limited and compromises the relevance of the studies. The authors should evaluate the effects of the K186A mutation on gamma secretase

kinetics to answer whether the substrate dynamics affect substrate recognition (K_m)? or rate by facilitating helix unwinding?

A: We appreciate the reviewer's specific comments and request regarding a deeper functional evaluation of our findings. Unfortunately, since we perform our cleavage assay with membranes and not with in vitro purified protein components, exact measurements of K_m and rates are not possible in this setup, since substrate concentration cannot be varied. Thus, while we agree that it would be interesting to know at which level TREM2 substrate dynamics influences the overall cleavage process, this remains to be answered in future studies.

Furthermore, it would be important to demonstrate that at least another mutation affecting the dynamics of the TREM-2 TMH (such as the K186P) has similar functional effects on the gamma secretase endopeptidase. Otherwise, the authors cannot rule out in their model another specific role of the lysine residue at the position 186 of TREM2. Generating the above described data is feasible, will help significantly with the interpretation of the structural data and support better the model presented in the manuscript.

A: We now added two additional TREM2 mutations to our analysis, K186L and K186P, as suggested by the reviewer and analyzed both variants with NMR and in our in vitro γ -secretase cleavage assay (see new Fig. 5 and new Figs. EV 7&9). We now demonstrate that K186L is behaving very similar to the K186A variant, both, in respect to its helix stabilizing effect as well as in regard to the shift of the predominant ϵ -cleavage site to amino acid 195. In contrast, K186P, where the proline residue leads to the induction of a kink in the helix and thus to a helix destabilization, retains a similar cleavage pattern like the wt TREM2 TMH (see new Fig. 5 and new Fig. EV 7&9). These data therefore establish that structural dynamics of TREM2 at the C-terminal end of the TMH determine the site where the initial ϵ -cleavage takes place. These findings are also consistent with our observation that the K186P decreases helicity whereas K186A and K186L increase helicity of the TMH (see new Fig. EV 7f and g).

2) Interestingly, MD simulations and paramagnetic relaxation enhancement (PRE) experiments of wild-type TREM2-TMH indicate that the C-terminal part of the predicted TMH is positioned outside the membrane. In contrast the drop in the intensity ratio within the entire K186A mutant TMH (residues 173-196) indicate a stable membrane-embedded structure. The observation that the part of the substrate that is actually cut by γ -secretase is not embedded in the membrane is unprecedented, the origin is well documented and discussed in the manuscript but the implications for the γ -secretase substrate recognition/cleavage mechanisms are not discussed. The authors should elaborate on the potential implications of this finding.

A: The kink region of TREM2-TMH is indeed oriented toward the membrane surface in the lipid head group region. We now added specific comments on the potential implication for recognition and cleavage by γ -secretase. In the Discussion we extended our view of how γ -secretase may sample the TREM2 TMH for unstructured regions when it has reached the substrate-binding groove of γ -secretase. As the transition of a substrate from the lipid bilayer to the enzyme is currently not understood, we felt, however, that a more detailed discussion of how the TREM2 TMH cleavage region re-positions from the membrane surface to the active site would be too speculative at the moment. In general, γ -secretase substrate cleavage at a surface exposed position has been reported previously with E-cadherin (Marambaud *et al*, *EMBO J*, 2002). In that case high intrinsic TMH stability interferes with unwinding of the substrate by γ -secretase leading to cleavage at a surface exposed position. This concept is in line with our findings with TREM2, where TMH stabilization results in a shift of the cleavage site to a more C-terminal position, and has been added in the Discussion.

3) The chemically induced unfolding of TREM2-TMH wild-type (Fig. 6d) and the K186A variant (Fig. 6e)

shows that the less dynamic variant is more stable than the wild-type protein. This reviewer wonders if the different stabilities of the wild type vs. mutant substrate have an impact on the stability of gamma-secretase-substrate interactions. It has been recently shown that AD-linked mutations impair the stability of gamma-secretase-substrate interactions and the effect shifts amyloid beta product length (Szaruga et al, Cell 2017). It would be very interesting to analyze how the substrate stability affects the stability of the gamma-secretase-TREM2 complex.

A: Thanks to the reviewer for this excellent comment. It would be very interesting to see direct correlations between substrate stability and the affinity with γ -secretase. However, as outlined already above regarding the reviewer's point on kinetic studies, we do not have the assays at hand that would allow measuring substrate-enzyme complex stabilities. Again, we believe that these questions, while interesting, are beyond the scope of this initial study, that aimed at linking conformational dynamics of the TREM2 TMH with the sites of the initial ϵ -like γ -secretase cleavage. The questions raised by the reviewer should thus be addressed in follow up-studies that focus more specifically on these aspects.

Minor points:

1) Several references do not refer to the original research papers but to review articles. For example: ' γ -Secretase (Wolfe, 2019), then generates A β from the membrane-retained APP C-terminal fragment (CTF) by a number of sequential, approximately 3 amino acids spaced cleavages, initiated at the so-called ϵ -cleavage site located very close to the C-terminal end of the TMH (Steiner, Fukumori et al., 2018).'

To improve clarity, this reviewer suggests to refer to the original articles or indicate by 'reviewed/discussed in' that the used reference is a review article.

A: Done

2) Spaces are missing in various parts of the manuscript. Here some examples:

'which subsequently undergoes regulated sequential intramembrane proteolysis(Wunderlich, Glebov et al., 2013) (Fig 1a).'

'and Notch1 (Deatherage, Lu et al., 2017)are shorter than for TREM2.'

'1 mL of 600 μ M MSP1D1 Δ H5(Hagn et al., 2013, Hagn et al., 2018)'

'describing a two-state folding model(Privalov, 1979).'

A: Done

3) Referencing problem:

(Ref (Morrisett, David et al., 1973))

A: Done

4) In the introduction a period is missing after: "Complete degradation of the CTF is crucial to prevent inhibition of DAP12 that would encompass productive full-length-dependent TREM2 signaling. (Wunderlich et al., 2013) ".

A: Done

Referee #3:

In this paper NMR is used to determine the solution structure of a transmembrane substrate of the γ -secretase, TREM2. Variations in the TREM place a person in a higher risk category for neurodegenerative disease. Typically, substrates of intramembrane proteases consist of transmembrane segments, which must be unwound prior to cleavage. For γ -secretase over 80 substrates exist. Recent structures demonstrate that substrates unwind and form β -sheet with the enzyme. It is not known how γ -

secretase selects its substrates and the authors propose intrinsic dynamics of the substrate may influence recognitions. This study set out to address this by studying the triggering receptor expressed on myeloid cells (TREM) 2 as a substrate and examination of its structure and cleavage. The NMR solution structure revealed a helix segment with a less ordered region near a charged Lysine residue, in other words a kink followed by a helical section that was proposed to influence dynamics. To eliminate the possibility of the influence of detergent, the structure was confirmed with the membrane protein in nanodisks. Membrane thickness was also ruled out in influencing this kink. The helical structure was stabilized upon interaction with its physiological partner DAP12. Mutation of this charge residue to Alanine resulted in changes in the chemical shift, suggestive of a continuous helix, with similar chemical shifts (i.e. stabilization) when in association to DAP12. A structural model was generated to show TREM and DAP interact with each other via a salt-bridge (utilizing the Lys). NOEs were obtained to measure dynamics and the K186A mutation resulted in less dynamics in this region. MD simulations in lipid bilayers suggest the Lys residues causes the helix to tilt in the membrane via associations with the phosphate headgroups. Paramagnetic relaxation enhancement (PRE) experiments were used to show the C-terminal region is outside of the hydrophobic area, while the K186A is entirely in the micelle. This was conducted in DPC micelles. A cell-free in vitro -secretase assay was used to measure release of the C-terminal fragment. They show this using this assay in conjunction with mass spectrometry that the mutation K186A results in a shift in cleavage. A structural model for the different recognition is provided. A novel hypothesis is put forth based on the fact that dynamic regions is one mechanism that influences substrate recognition for gamma secretase.

Overall this is a very well written manuscript. The concept that structural dynamic regions play a role in substrate recognition for g-secretase substrate is interesting given that there is no sequence conservation for the numerous substrates. The data present in the manuscript is convincing that changes in the single Lys residue alter the structural nature and cleavage of the TREM12 substrate.

A: Thanks to the reviewer for their positive summary

Major points:

Is the Lys involved in a salt bridge should its mutation affect interaction with DAP12? We see that mutation of the Asp residue in DAP leads to a loss in chemical shift upon interaction. Did the L186A also have the same effect? This is clear from the figures but the results section does not make this point clearly. i.e. on Page 7.

A: We now included new NMR data in Fig. EV 3 where we used isotope-labeled TREM2-TMH K186A and added unlabeled DAP12. With this pair of proteins, no interaction could be detected, just as with DAP12 D50A and wt TREM2-TMH. This is in line with the previously published findings that homozygous mutations of K186 lead to Nasu-Hakola-disease since they disrupt binding to DAP12 and therefore resemble a full loss of function.

On page 10 the following statement: This pattern suggests that the first helix is indeed a transmembrane helix whereas the kink region and the C-terminal helix are located outside the membrane, should be adjusted to state In DPC micelles. There is no evidence that this is outside in the physiological cellular environment which would have thicker membranes compared to DPC.

A: Has been rephrased as follows:

"This pattern suggests that the first helix is indeed a transmembrane helix whereas the kink region and the C-terminal helix are located at the surface of the DPC micelle.

The hypothesis that dynamic flexibility is one mechanism influencing substrate is interesting however in the discussion I would like to see if this is the case for any of the other 80 plus known substrates for g-secretase. Only APP HDX and simulation experiments are discussed.

A: We added more references with examples as well as theoretical predictions of helix dynamics (with Notch1, ErbB4, together with APP and the TREM2 variants, see new Fig. EV 8), where we can clearly see that cleavage is taking place at positions close to the C-terminus of the TMH (~3

residues away from the unstructured region) that also show reduced rigidity as compared to the more central part of the TMH, along with the two references to the recent cryo-EM structures with APP and Notch1, where the C-terminal regions were described to undergo a helix-to- β -strand transformation. These helpful insights fit very well to the data we obtained with TREM2. The cleavage site in TREM2-wt is 6 residues away from the C-terminus of the helical regions (2nd shorter helix), which would be energetically highly unfavorable for γ -secretase to unwind such a long stretch. However, the presence of the charged K186 residue in the transmembrane region leads to a destabilization of the helical secondary structure that facilitates cleavage at the 192-193 position. In the TREM2-K186A variant, where this helix defect is not present any more, the cleavage site is again similar to many other γ -secretase substrates (3 residues away from the C-terminal end of the TMH). Thus, we can conclude that γ -secretase cleavage might follow general principles that are in part governed by substrate stability. However, detailed knowledge can only be obtained by further experimental studies on the intrinsic dynamics of a larger number of substrate helices.

Minor points to address:

1. Sigma protease inhibitor mix is mentioned several times in the methods, however several exist at Sigma. Please specify precise product, especially since the research is protease focused.

A: Has been corrected

Dr. Franz Hagn
TUM
Department of Chemistry
Lichtenbergstrasse 4
Garching, Bavaria 85747
Germany

24th Jul 2020

Re: EMBOJ-2019-104247R
Intramembrane cleavage of TREM2 is determined by its intrinsic structural dynamics

Dear Franz,

Thank you for submitting your revised manuscript to The EMBO Journal. It has now been assessed once more by two of the original referees, both of whom are fully satisfied with the revisions. I am therefore happy to inform you that we would like to proceed further with publication of this work, following final modifications to incorporate various editorial points as detailed below.

- Pre-acceptance checks have raised a few queries with the data descriptors in the main and EV figure legends, which you will find as comments in the attached edited/commented Word documents with activated "Track changes" option. I would appreciate if you incorporated the requested final text modifications and answered the Figure legend queries directly in this version (and modified figures where necessary), uploading the edited main text document upon resubmission with changes/additions still highlighted via the "Track changes" option, to facilitate our final checking.
- please upload all main Figures and all Expanded View figures (which have been removed from the Word file) as individual files with sufficient resolution/quality for production; the text file should only contain text/legends/tables.
- While we can accommodate more than 6 main figures, EV Figures are usually limited to 5-6. Therefore, please consider re-arranging, either by promoting some of the currently 9 EV figures to main figures (if justified), or by moving some (or all) EV figures into an Appendix PDF (in this case, their legends should go into the Appendix file too, together with a brief table of content for the Appendix. Figures and references to them would have to be renamed "Appendix Fig S1/2/3..."). For more information please refer to embopress.org/page/journal/14602075/authorguide#expandedview
- Please include database hyperlinks in the "Data Availability" section (suggested wording: "The [structural coordinates | microarray | mass spectrometry] data from this publication have been deposited to the [name of the database] database [URL] and assigned the identifier [accession | permalink | hashtag].")
- To ensure broad visibility and accessibility, I wonder whether the title could be made slightly more explicit, by providing a bit more context information? Maybe by reference to AD risk factor, or to gamma-secretase?

- Finally, please provide suggestions for 3-5 one-sentence 'bullet points', containing brief factual statements that summarize key aspects of the paper - they will form the basis of an editor-written 'Synopsis' accompanying the online version of the article. Please see the latest research articles on our website (embojournal.org) for examples. Please also provide a simplified schematic image for the online synopsis, maybe base on an exemplary structural snapshot; it is important to keep in mind that such a synopsis image should be in landscape format and restricted to a rather modest size, 550 pixels in width and maximum 400 pixels in height.

I am therefore returning the manuscript to you for a final round of minor revision, to allow you to make these adjustments and upload all modified files. Once we will have received them, we should be ready to proceed with formal acceptance and production of the manuscript.

With kind regards,

Hartmut

Hartmut Vodermaier, PhD
Senior Editor / The EMBO Journal
h.vodermaier@embojournal.org

- a point-by-point response to the referees' comments, with a detailed description of the changes made (as a word file).
- a word file of the manuscript text.
- individual production quality figure files (one file per figure)
- a complete author checklist, which you can download from our author guidelines (<https://www.embopress.org/page/journal/14602075/authorguide>).
- Expanded View files, replacing Supplementary Information (Please see <https://www.embopress.org/page/journal/14602075/authorguide#expandedview>)

Please remember: Digital image enhancement is acceptable practice, as long as it accurately represents the original data and conforms to community standards. If a figure has been subjected to significant electronic manipulation, this must be noted in the figure legend or in the 'Materials and

Methods' section. The editors reserve the right to request original versions of figures and the original images that were used to assemble the figure.

Revision to The EMBO Journal should be submitted online within 90 days, unless an extension has been requested and approved by the editor; please click on the link below to submit the revision online before 22nd Oct 2020:
<https://emboj.msubmit.net/cgi-bin/main.plex>

Referee #2:

The authors have addressed the most important points and I am satisfied with the overall revision of the manuscript. I consider that the paper is ready for publication in the EMBO Journal.

Referee #3:

The authors have addressed all concerns expressed in my first review. The paper is now suitable for publication. I have no remaining concerns.

Dear Hartmut,

Thank you for handling our submission. We are very happy about the positive outcome of the review process (comments below) of our manuscript and are glad that the reviewers are now fully satisfied with the current version of the manuscript.

Attached please find the final version of the manuscript where the requested changes have been made (with the “track changes” option active). Furthermore, we uploaded all figure files as high-resolution images and provided a 5-bullet-point list for the editorial summary together with a summarizing image. In order to shorten the EV part of the manuscript, we created an Appendix containing two figures.

Thanks again and we are looking forward to your final acceptance message in due course.

Kind regards,
Franz

Referee #2:

The authors have addressed the most important points and I am satisfied with the overall revision of the manuscript. I consider that the paper is ready for publication in the EMBO Journal.

Referee #3:

The authors have addressed all concerns expressed in my first review. The paper is now suitable for publication. I have no remaining concerns.

Authors' response: Thanks to all three reviewers for their constructive comments that helped us to improve the significance of the paper.

Dr. Franz Hagn
TUM
Department of Chemistry
Lichtenbergstrasse 4
Garching, Bavaria 85747
Germany

29th Jul 2020

Re: EMBOJ-2019-104247R1

β -secretase cleavage of the Alzheimer risk factor TREM2 is determined by its intrinsic structural dynamics

Dear Franz,

Thank you for submitting your final revised manuscript for our consideration. I am pleased to inform you that we have now accepted it for publication in The EMBO Journal.

Your article will be processed for publication in The EMBO Journal by EMBO Press and Wiley, who will contact you with further information regarding production/publication procedures and license requirements. You will also be provided with page proofs after copy-editing and typesetting of main manuscript and expanded view figure files.

Should you be planning a Press Release on your article, please get in contact with embojournal@wiley.com as early as possible, in order to coordinate publication and release dates.

Yours sincerely,

Hartmut

Hartmut Vodermaier, PhD
Senior Editor / The EMBO Journal
h.vodermaier@embojournal.org

Please note that it is EMBO Journal policy for the transcript of the editorial process (containing referee reports and your response letter) to be published as an online supplement to each paper. If you do NOT want this, you will need to inform the Editorial Office via email immediately. More information is available here:

Corresponding Author Name: Franz Hagn

Journal Submitted to: EMBO J

Manuscript Number: EMBOJ-2019-104247R